



# Evaluation of modelled climatologies of $O_3$, CO, water vapour and $NO_y$ in the upper troposphere–lower stratosphere using regular in situ observations by passenger aircraft

Yann Cohen[1], Didier Hauglustaine[1], Bastien Sauvage[2], Susanne Rohs[3], Patrick Konjari[4],
Ulrich Bundke[3], Andreas Petzold[3], Valérie Thouret[2], Andreas Zahn[5], and Helmut Ziereis[6]

[1]Laboratoire des Sciences du Climat et de l'Environnement, LSCE-IPSL (CEA-CNRS-UVSQ), Université Paris-Saclay, 91191 Gif-sur-Yvette, France

[2]Laboratoire d'Aérologie, Université de Toulouse, CNRS, UPS, France

[3]Forschungszentrum Jülich GmbH, Institute of Energy and Climate Research 8 – Troposphere, Jülich, Germany

[4]Forschungszentrum Jülich GmbH, Institute of Energy and Climate Research 7 – Stratosphere, Jülich, Germany

[5]Institute of Meteorology and Climate Research, Karlsruhe Institute of Technology, Karlsruhe, Germany

[6]The German Aerospace Center (Deutsches Zentrum für Luft- und Raumfahrt; DLR), Institute for Atmospheric Physics

**Correspondence:** yann.cohen.09@gmail.com

**Abstract.** Evaluating the global chemistry models in the upper troposphere–lower stratosphere (UTLS) is an important step toward a further understanding of its chemical composition. The latter is regularly sampled through in situ measurements based on passenger aircraft, in the framework of the In-service Aircraft for a Global Observing System (IAGOS) research infrastructure. This study focuses on the comparison of the IAGOS measurements in ozone, carbon monoxide (CO), nitrogen

5  reactive species ($NO_y$) and water vapour, with a 25-year simulation output from the LMDZ-OR-INCA chemistry-climate model. For this purpose, we present and apply an extension of the Interpol-IAGOS software that projects the IAGOS data onto any model grid, in order to derive a gridded IAGOS product and a masked model product that are directly comparable to one another. Climatologies are calculated in the upper troposphere (UT) and in the lower stratosphere (LS) separately, but also in the UTLS as a whole, as a demonstration for the models that do not sort out the physical variables necessary to distinguish between the UT and the LS. In the northern extratropics, the comparison in the UTLS layer suggests that the geographical distribution

10  in the tropopause height is well reproduced by the model. In the separated layers, the model simulates well the water vapour climatologies in the UT, and the ozone climatologies in the LS. The opposite biases in CO in both UT and LS suggest that



the cross-tropopause transport is overestimated. The NO$_y$ observations highlight the difficulty of the model in parameterizing the lightning emissions. In the tropics, the upper-tropospheric climatologies are remarkably well simulated for water vapour, as the observed CO peaks due to biomass burning in the most convective systems, and the ozone latitudinal variations. Ozone is more sensitive to lightning emissions than to biomass burning emissions, whereas the CO sensitivity to biomass burning emissions strongly depends on the location and on the season. Through this evaluation, the present study demonstrates that the Interpol-IAGOS software is a tool facilitating the assessment of the global model simulations in the UTLS, potentially useful for any modelling experiment involving chemistry-climate and chemistry-transport models.

## 1 Introduction

The upper troposphere–lower stratosphere (UTLS) is defined as a thin transition layer around the tropopause. It is a key region regarding the chemical composition in both the troposphere and the stratosphere, acting as a complex transport barrier (Gettelman et al., 2011) with a varying strength (e.g. Zhang et al., 2019). The UTLS is also a relevant altitude domain in matter of radiative forcing (Riese et al., 2012) from ozone (O$_3$) and water vapour (denoted here as H$_2$O), two species classified amongst the most important greenhouse gases (Arias et al., 2021; Szopa et al., 2021). Furthermore, both play an important role in the atmospheric composition: in the stratosphere, ozone absorbs most of energetic ultraviolet radiation whereas water vapour acts as an ozone sink through catalytic cycles; in the troposphere, their combined presence changes the air oxidizing capacity by generating hydroxyl radical (OH). In the upper troposphere (UT), water vapour is also a key species regarding the cirrus clouds formation and life cycle, whose high radiative forcing is still of large uncertainty (Krämer et al., 2020). Carbon monoxide (CO) is one of the main tropospheric ozone precursors and the main sink for OH (Lelieveld et al., 2016), such that its oxidation competes with methane (CH$_4$) chemical destruction, thus increasing the latter's lifetime. Nitrogen oxides (NO$_x$) are an O$_3$ sink in the stratosphere but a necessary ingredient for tropospheric O$_3$ formation, with an important contribution in the free troposphere (e.g. Sauvage et al., 2007a; Grewe et al., 2012). All these gases are thus classified amongst the essential climate variables (Bojinski et al., 2014). NO$_x$ are converted back and forth into their reservoir species (NO$_z$), making the ensemble of the nitrogen reactive species (NO$_y$ = NO$_x$ + NO$_z$) a relevant variable for understanding photochemical processes.

Chemistry-climate models (CCMs) and chemistry-transport models (CTMs) are essential tools for calculating budgets for individual chemical species with their radiative forcings since the beginning of the industrial period (e.g. Eyring et al., 2013; Collins et al., 2017), for understanding their sources and sinks, and for predicting the evolution of the atmosphere through the current century. Assessing the UTLS chemical composition in global simulations covering the last decades is a relevant step towards reducing the uncertainties on several dynamical processes. As CO is emitted mostly at the surface and as its lifetime is



sufficiently long to be transported up to the UTLS (e.g. Lelieveld et al., 2016), it can be used to assess convection in the models. $NO_y$ also provide information on moist convection, since lightning is the major source of $NO_x$ in the free troposphere (Allen et al., 2010; Cooper et al., 2009), thus an important source of $NO_y$ (Gressent et al., 2014). Since the stratosphere is particularly rich in nitric acid ($HNO_3$) because of nitrous oxide ($N_2O$) chemical destruction, the $NO_y$ can also provide information on air

masses origins in the extratropical lower stratosphere (Popp et al., 2009). As $H_2O$ and CO on one hand and $O_3$ and $NO_y$ on the other hand are more abundant respectively in the troposphere and the stratosphere, these four tracers are useful in evaluating stratosphere-troposphere exchange.

 Assessing CCM or CTM simulations relies on the comparison with observational data sets. However, in matter of vertical resolution, few of them are suited for diagnosing the UTLS status, and fewer again to account for the UTLS vertical hetero-

geneity. LiDAR (Light Detection And Ranging) instruments provide notably $O_3$ measurements with vertical resolutions of $\sim 1$ km or less near the tropopause (Gaudel et al., 2015a; Granados-Muñoz and Leblanc, 2016), and can be used with *in situ* measurements performed by ozonesondes. Although both provide vertical profiles through a large-scale network in their ensemble, they cover areas limited to the vicinity of ground stations. *In situ* measurements are also provided by aircraft campaigns up to 20 km above sea level, highlighting small-scale events inaccessible for most model resolutions (Hegglin et al., 2004), or the

need to improve some parameterizations (e.g. regarding $NO_y$: Brunner et al., 2005) but they are too sparse in space and time to derive long-term statistics.

 In situ measurements on board commercial aircraft provide a frequent and large-scale sampling at the cruise altitudes (9– 12 km). Based on these observations, several scientific programs allowed to highlight large-scale features since the 1970s, as TROZ (TRopospheric OZone: Fabian and Pruchniewicz, 1977), GASP (Global Atmospheric Sampling Program: Falconer

and Holdeman, 1976) and more recently NOXAR (Nitrogen OXides and ozone along Air Routes: Brunner et al., 1998; Dias-Lalcaca et al., 1998), with an observation period spreading over four years or less.

 Since more than two decades, the In-service Aircraft for a Global Observing System research infrastructure (IAGOS: Petzold et al., 2015) provides regular aircraft measurements simultaneously for ozone, water vapour, CO and, to a lesser extent, $NO_y$. The measurements recorded during the cruise phases now compose a long-term data set with a high vertical resolution in the

UTLS and a wide geographical coverage, especially in the northern mid-latitudes. Amongst the applications involving model evaluations, Law et al. (2000) used the IAGOS-MOZAIC data from 1994 until 1996 to assess a set of models in the UTLS. Brunner et al. (2003) combined the first four years of IAGOS-MOZAIC measurements with two aircraft campaigns for a similar purpose. But in the end, few model assessments took benefit of the whole IAGOS database. Several studies used the IAGOS database over a long period, but on a regional scale only, for instance to evaluate the MACC (Monitoring Atmospheric

Chemistry and Climate) reanalysis over Europe (Gaudel et al., 2015b), the Community Earth System Model CAM4-chem
    (Community Atmospheric Model, version 4: Tilmes et al., 2016) over the Narita airport (Japan), or the GEOS-Chem (Goddard
    Earth Observing System) model over the Indian subcontinent (David et al., 2019).

    More recently, Cohen et al. (2021) developed the Interpol-IAGOS software based on the whole cruise IAGOS data set to
    assess part of a reference experiment (so-called REF-C1SD), in the framework of the Chemistry-Climate Model Initiative
(CCMI: Eyring et al., 2013) program. A first application was made on the MOCAGE CTM (MOdélisation de la Chimie
    Atmosphérique à Grande Échelle: Guth et al., 2016) using ozone and CO measurements during 1995–2013 and 2002–2013
    respectively, and was partly based on the use of the model potential vorticity (PV) field to separate the upper troposphere
    (UT) and the lower stratosphere (LS). However, the software was designed for multi-model comparisons that required the
    outputs to be archived in monthly means, leading to a low resolution in the UT and LS definitions. Along with providing an
estimation of the impact from lightning and biomass burning on the UTLS chemical composition using the LMDZ-OR-INCA
    model, the present study consists of going further into the development and the application of the methodology presented in
    Cohen et al. (2021), following three major improvements. First, the daily resolution of the current simulation allows a more
    accurate separation between UT and LS. Second, the anthropogenic emissions have a monthly resolution, thus allowing a better
    comparison than in the previous study. Third, the comparison now involves $O_3$, CO but also $H_2O$ measurements on decadal
timescales, as well as $NO_y$ measurements. The latter are substantially less frequent, so we merged the IAGOS-MOZAIC and
    the IAGOS-CARIBIC data sets in order to compensate this lack of data as much as possible. In Sect. 2, we describe the
    IAGOS data set, the LMDZ-OR-INCA model, the simulation setup and the method used to process the data and to assess the
    simulation. In Sect. 3, we apply the methodology to the assessment of a bi-decadal simulation from the LMDZ-OR-INCA
    CCM. We finally discuss the contribution of lightning and biomass burning on the modelled chemical fields. The last two steps
treat the extratropical and tropical latitudes separately, in order to account for the differences in the seasons definitions and in
    the mean tropopause altitude.

## 2  Materials and methods

### 2.1  IAGOS observations

The IAGOS research infrastructure (www.iagos.org) provides *in situ* measurements of chemical species on board several com-
mercial aircraft. Its predecessors MOZAIC (Measurements of water vapor and OZone by Airbus In-service airCraft: Marenco
    et al., 1998) and CARIBIC (Civil Aircraft for the Regular Investigation Based on an Instrument Container: Brenninkmeijer



et al., 1999, 2007; Stratmann et al., 2016) relied on the same principle. Hence, their approaches are complementary. MOZAIC started with a fleet of five equipped aircraft measuring ozone and water vapour since August 1994. CO measurements started in December 2001 and NO$_y$ measurement were operational on one aircraft between April 2001 and May 2005. On the other hand, CARIBIC samples a wide variety of atmospheric species since 1997, including the ones measured by MOZAIC, from one single aircraft. Since the merge of the two programs in 2008, their respective databases are referred as IAGOS-Core and IAGOS-CARIBIC. In the present study, we consider them as a single database called IAGOS hereafter, approach validated by Blot et al. (2021) for ozone and CO. The period we are analysing spreads from Aug. 1994 until Dec. 2017.

In IAGOS-Core, ozone (CO) is measured with a ultraviolet (infrared) absorption spectrometer, whereas water vapour is sampled with a capacitive hygrometer, and NO$_y$ with a chemiluminescence gold converter. Respectively, their accuracy, precision, and time response are 2 ppb, 2 % and 4 s for ozone (Thouret et al., 1998); 5 ppb, 5 % and 30 s for CO (Nédélec et al., 2003; Nédélec et al., 2015); 5 % relative humidity with respect to liquid water (RHL) and 5–300 s for water vapour (Helten et al., 1998; Neis et al., 2015a, b) or 6 % RHL in the thermal tropopause at mid-latitudes (Smit et al., 2014); 50 ppt, 5 % and 4 s for NO$_y$ (Volz-Thomas et al., 2005; Pätz et al., 2006). Concerning water vapour, a potential drift of the sensor baseline during long deployment periods is corrected by applying the so-called in-flight calibration (IFC), which uses flight sequences in very dry conditions to determine the offset at zero relative humidity (Smit et al., 2008). The validity range of the humidity sensor spreads between 5 and 70 % RHL (Neis et al., 2015a).

In IAGOS-CARIBIC, ozone (O$_3$) is measured with a combination of a dry chemiluminescence detector and a UV absorption spectrometer (vacuum UV fluorescence). Water vapour measurements are performed with a photoacoustic laser spectrometer and a frost-point hygrometer, and NO$_y$ with a chemiluminescence gold converter again. Accuracy, precision, and time response are listed respectively as follow: 0.5 ppb or 1 % and 4 s for ozone in the case of UV absorption, or 0.2 s in the case of chemiluminescence (Zahn et al., 2012); less than 2 ppb, 1–2 ppb and 2 s for CO (Scharffe et al., 2012); less than 1 ppm, less than 3 % and 4–20 s for water vapour in the case of the laser photoacoustic spectrometer, or 5–90 s in the case of frost-point hygrometer (Zahn et al., 2014; Dyroff et al., 2015); 6.5–8 % and 1 s for NO$_y$ (Ziereis et al., 2000; Stratmann et al., 2016).

## 2.2 The LMDZ-OR-INCA model

The LMDZ-OR-INCA global chemistry-aerosol-climate model results from the on-line coupling between the LMDZ general circulation model (Laboratoire de Météorologie Dynamique, version 6: Hourdin et al., 2006) and the INCA model (INteraction with Chemistry and Aerosols, version 5: Hauglustaine et al., 2004). The coupling between LMDZ and the ORCHIDEE dynamical vegetation model (Krinner et al., 2005) ensures the interaction between the atmosphere and the land surface. The current



configuration is characterized by a vertical grid extending up to 70 km, discretized into 39 hybrid levels. The horizontal grid

cells spread over $1.25°$ in latitude and $2.5°$ in longitude. The primitive equations in the general circulation model (GCM) are

solved with a 3 min time-step, large-scale transport of tracers is carried out every 15 min, and physical and chemical processes

are calculated at a 30 min time interval. Further detail on the GCM is provided in Hourdin et al. (2006).

    The INCA model first included a state-of-the-art $CH_4$-$NO_x$-CO-NMHC-$O_3$ tropospheric photochemistry (Hauglustaine

et al., 2004; Folberth et al., 2006). In this model version, the tropospheric photochemistry and aerosols scheme gathers a

total of 123 tracers including 22 aerosol tracers. The model comprises 234 homogeneous chemical reactions, 43 photolytic

reactions and 30 heterogeneous reactions. The gas-phase version has been extensively compared to observations around the

tropopause region. Aerosols are both represented in species with anthropogenic sources such as sulfates, nitrates, black carbon,

particulate organic matter, and natural species such as sea salt and dust. The processes involving ammonia and nitrate aerosols

are described in Hauglustaine et al. (2014). The INCA model has been recently extended to include an interactive chemistry in

the stratosphere and mesosphere, and now includes chemical species and reactions specific to the middle atmosphere. A total

of 31 species were added to the standard chemical scheme, mostly belonging to the chlorine and bromine chemistry, and 66

gas-phase reactions and 26 photolytic reactions (Terrenoire et al., 2022; Pletzer et al., 2022).

    In this study, the LMDZ GCM surface zonal and meridional wind components are nudged towards the meteorological data

from the European Center for Medium-Range Weather Forecasts (ECMWF) ERA-Interim reanalysis, with a relaxation time of

2.5 h (Hauglustaine et al., 2004). The ECMWF fields are provided every 6 hours and interpolated onto the GCM grid.

    The historical global anthropogenic emissions are taken from the Community Emissions Data System inventories (CEDS:

Hoesly et al., 2018) up to 2014, followed by the projections based on Gidden et al. (2019). Concerning China, the anthro-

pogenic emission inventories are replaced by the Zheng et al. (2018) emissions available for the period 2010–2017. The global

biomass burning emissions are taken from van Marle et al. (2017) up to 2015, followed by the projections from Gidden et al.

(2019) as for anthropogenic emissions. The biogenic surface fluxes of isoprene, terpenes, methanol and acetone as well as

NO soil emissions have been calculated off-line by the ORCHIDEE vegetation model as described in (Messina et al., 2016).

The lightning $NO_x$ parameterization is described in Jourdain and Hauglustaine (2001). The lightning frequency follows the

parameterization from Price and Rind (1992). In this simulation, a rescaling constrains the mean global flash rate at 46.3 flash

$yr^{-1}$, consistently with the annual climatologies derived from both Lightning Imaging Sensor and Optical Transient Detector

(LIS–OTD) satellite instruments in Cecil et al. (2014), from 1995 until 2010. This rescaling accounts for the different LIS

and OTD sampled latitude bands, and for their different sampling periods. The lightning $NO_x$ ($LNO_x$) emissions are then

redistributed vertically, based on Ott et al. (2010).



In order to enhance the understanding of both the simulation biases and the well-reproduced features, the run presented here
has been repeated once without lightning emissions and once without biomass burning emissions. Hereafter, we refer to these
simulations with the "-no-LNO$_x$" and "-no-BB" suffixes respectively. In order to complete information regarding ozone, we
added the stratospheric ozone tracer (O$_3$S) and the inert-stratospheric ozone tracer (O$_3$I). Both refer to ozone originating from
the stratosphere, but the latter is destroyed by dry deposition only, whereas O$_3$S is destroyed by chemical reactions as well,
thus with the same lifetime as tropospheric ozone.

## 2.3 Building up the new gridded IAGOS product

### 2.3.1 Data projection onto the model grid

The strategy consists of adapting the IAGOS data to the studied simulation in matter of spatial resolution, following a linear
reverse interpolation. As illustrated in Fig. 1 in Cohen et al. (2021), for a given month, each measurement point is projected
onto its adjacent grid cells, where a normalized weight is assigned depending on the distance from the measurement point.
For a given grid cell, a monthly mean value is then derived from a weighted averaging between the projections from all the
neighbouring measurement points onto the grid cell. For filtering purposes, an equivalent sample size N$_{eq}$ is also provided by
summing up all these weights. This IAGOS product is therefore called IAGOS-DM-INCA, the -DM first suffix referring to
the distribution onto the model grid, and the -INCA second suffix precising the destination model. Since there is no multi-
model comparison in the current paper, we simply call it IAGOS-DM hereafter. In order to derive a comparable product from
the simulation, the daily model outputs are also averaged over the months, filtering out the days without measurements. The
subsequent product is named INCA-M hereafter, the -M suffix referring to the mask with respect to the IAGOS sampling.

### 2.3.2 Separation between UT and LS

Diagnosing the UTLS chemical behaviour in detail requires the differentiation between UT and LS. This is why the projections
described above can optionally involve the model potential vorticity (PV) field in order to locate the dynamical tropopause,
defined as PV$_{TP}$ = 2 potential vorticity units (PVU) in Thouret et al. (2006). According to the same study, the tropopause is
represented as a transition layer excluded from both troposphere and stratosphere, which ensures the UT and the LS to be
sufficiently isolated from each other. As in Cohen et al. (2021), the LS is represented by all the sampled grid points where the
PV exceeds 3 PVU, keeping in mind that the commercial aircraft usually do not fly above 12 km. Concerning the UT, a sampled
grid point is considered as upper-tropospheric if its PV is lower than 2 PVU, if it is not adjacent to the 2 PVU isosurface, and
if its hybrid $\sigma$-pressure value is lesser than 400 hPa. The second condition enhances the isolation of the UT from the mixing



zone. Last, in order to assess the model ability in reproducing the chemical composition in both layers without influence from errors in the PV field, we fix another filtering condition based on ozone measurements. According to Cohen et al. (2021), an upper-tropospheric (lower-stratospheric) daily grid point is filtered out when its observed ozone mean value is greater (lesser) than 140 (60) ppb.

Since RHL values below 5 % are outside the measurement range of the IAGOS-Core water vapour sensor and tend to be measured with a wet bias, we apply an additional filter that consists of masking the daily grid points with more than 20 % of the measurements drier than 10 % RHL. Such dry air masses are frequently encountered in the upper part of the LS (e.g. Zahn et al., 2014). Consequently, it is worth figuring out that the water vapour mean values derived in the LS are mostly representative of the lowermost part of this layer, contrary to the other measurements which are not concerned by this filter.

These very dry air masses are not present in the UT.

This study aims at presenting quasi horizontal maps and quantifying the mean gridpoint-to-gridpoint geographical variability, either for each season or for the whole year. It consists of the comparison between climatologies from IAGOS-DM and the simulation, both with and without an air mass discrimination. Consequently, a part of this software functionality does not need any PV field to be provided and is therefore accessible to every daily or monthly simulation output, for every global CCM and

CTM.

### 2.3.3 Deriving climatologies

The climatologies here refer to nearly horizontal maps derived from partial columns in the cruise altitudes. A time series of seasonal means is derived for each grid point, and then averaged throughout the years. Last, the mean yearly climatologies are defined as the average between the four seasonal climatologies. In the section dedicated to the tropics, zonal cross sections

are derived in the following zonal bands: 60° W–15° W, 5° W–30° E and 60° E–90° E. They correspond respectively to South America with the western Atlantic Ocean, Africa, and South Asia. Each area is defined as a compromise between sampling efficiency and spatial uniformity in the observed species, notably water vapour. The African zonal band is chosen as in Lannuque et al. (2021), as well as the division of the year into wet, dry and intermediate seasons. As the Intertropical Convergence Zone (ITCZ) behaviour varies between these regions, we reiterated the criteria used in Lannuque et al. (2021) to

adapt the seasons delimitation to the other regions. More precisely, we analysed month-by-month the mean zonal cross sections described by the observed zonal and meridian wind speeds, along with the water vapour mixing ratio, and gathered the months with the most similar features together. Notably, we focused on the stability in the location of the ITCZ, defined as a negative





**Table 1.** Characteristics of the chosen tropical regions.

| Region | Delimitation | Set of seasons |
|---|---|---|
| South America/Atlantic Ocean | 60° W–15° W | DJF–MAMJ–JA–SON |
| Africa | 5° W–30° E | DJFM–AM–JJASO–N |
| South Asia | 60° E–90° E | DJF–MAM–JJAS–ON |

minimum in the zonal wind speed, a weak meridian wind speed on average and a high water vapour mixing ratio. Table 1 synthesizes the definition of the regions and their associated sets of seasons.

### 2.3.4 Filtering conditions

We define the same filtering mechanism as done for $O_3$ and CO in Cohen et al. (2021). For a given species X at a latitude $\theta$, a long-term average on a grid cell is validated if the summed equivalent amount of data $N_{eq}$ reaches $N_{thres}(\theta, X) = N_{ref} f(\theta) g(X)$. $N_{ref}$ is a reference threshold for ozone. Following a sensitivity test, we chose it at 140 to optimize the robustness of the results against this threshold while limiting the loss of data. $f$ is a normalized function defined as $f(\theta) = \cos(\theta) / <\cos(\theta)>$, with $<\cos(\theta)>$ being the average of the cosine across the latitudes. The role of the $f(\theta)$ factor is to account for the grid cell area that decreases with latitude. g(X) is a factor depending on the X species measurement period $\Delta t_X$ and on the ratio R of equipped aircraft amongst the IAGOS fleet, such as $g(X) = R\Delta t_X / \Delta t_{O_3}$. By definition, R is set to 1 for $O_3$, CO and $H_2O$ and approximated at 1/6 for $NO_y$. The threshold is multiplied by a factor 4 for the yearly climatologies since every season is involved. In the tropics, the threshold is adapted to the seasons duration by applying a cross product. Last, the partial columns are derived by averaging across the vertical grid levels. They are validated if they represent at least two grid cells, in order to limit the biases linked to the mean measurement altitude that varies geographically.

### 2.3.5 Metrics used in the assessment

Without the separation between the UT and the LS, a given vertical grid level includes more stratospheric air masses in the mid-latitudes than in the subtropics. A simply averaged bias in $O_3$ (CO and $H_2O$) mean value and standard deviation would therefore be too dependent on biases in stratospheric (tropospheric) air composition. This inconvenience is fixed with the modified normalized mean bias (MNMB) and the fractional gross error (FGE), based on averages between relative mean biases. For a set of observed values $(o_i)_{i\in[\![1,N]\!]}$ and a set of simulated values $(m_i)_{i\in[\![1,N]\!]}$, this couple of metrics is defined as:

$$MNMB = \frac{2}{N} \sum_{i=1}^{N} \frac{m_i - o_i}{m_i + o_i} \qquad (1)$$



and

$$FGE = \frac{2}{N} \sum_{i=1}^{N} \left| \frac{m_i - o_i}{m_i + o_i} \right|$$   (2)

Consequently, a same relative bias for a poor-ozone and a rich-ozone air mass have the same weight in the resulting MNMB. From these definitions, and assuming that $m_i$ and $o_i$ are always positive, we can also derive the property:

$$|MNMB| \leq FGE \leq 2$$   (3)

The FGE thus represents a boundary for the MNMB. The MNMB absolute value equals the FGE when all the individual biases

$m_i - o_i$ have the same sign.

We use these metrics in the purpose of evaluating the reference simulation. It is not the case for the comparison with sensitivity simulations, since the normalizing factor in the MNMB definition varies from one simulation to another. In order to estimate explicitly the impact of lightning and biomass burning emissions, we choose to normalize the biases with respect to the observations only. Last, in any application, we systematically use the Pearson correlation coefficient defined as:

$$r = \frac{1}{N} \frac{\sum_{i=1}^{N} (m_i - \bar{m})(o_i - \bar{o})}{\sigma_m \sigma_o}$$   (4)

where $\bar{m}$ and $\bar{o}$ are the mean values and $\sigma_m$ and $\sigma_o$ their respective standard deviations.

## 3   Assessment of the simulated climatologies

### 3.1   Horizontal distributions

Ozone, CO, $NO_y$ and water vapour yearly distributions in the UTLS, UT and LS are shown in Figs. 1–4 respectively, and

their corresponding seasonal averages are available in Supplementary Material. Showing the results both with and without the separation is relevant because it can provide a better understanding for some biases visible in the UT or the LS. More generally, it is also relevant as a demonstration of the use of the Interpol-IAGOS software for both the simulations with and without an available potential vorticity field. Concerning the non-separated UTLS layer, it has to be noted that the vertical distribution of the IAGOS sampling relatively to the tropopause level varies geographically, resulting from both tropopause and cruise altitude

variations. Consequently, the values shown in the UTLS layer are not considered as representative of a geographically constant



vertical domain, and they do not necessarily represent the whole transition layer. Last, it must be kept in mind that the UTLS layer is not solely the merging of the UT and the LS, since it also comprises the vertical range between 2 and 3 PVU that separates the two layers.

Ozone climatologies (see Fig. 1) generally show geographical structures well reproduced by the model, i.e. the location of maxima in polar regions in the LS (west from Greenland and northern Siberia), the minimum on western equatorial Pacific Ocean in the UT, and the transition between subtropical and extratropical areas. In complement, the corresponding ozone seasonal climatologies available in Supplementary Material show that each point highlighted in this paragraph is representative of three seasons at least. Figure 2 highlights similarities between the CO climatologies from the two data sets, like the well model reproduction of the extreme values above the (sub)tropical convective and strongly emitting regions. However, one of the main features in the extra-tropical latitudes remains an important overestimation of CO in the LS characterized by a lessened geographical variability, and a moderate underestimation in the UT. The non-separated UTLS is relatively well reproduced in the mid-latitudes, with a moderate positive bias in the areas where the UT is not sampled, thus probably reflecting the lower-stratospheric positive bias. $NO_y$ are rather characterized by discrepancies between IAGOS-DM and INCA-M, especially in the UT with strong dipoles between positive and negative biases. The latter specificity is possibly an artifact due to the lower amount of measurements. Still, we identify collocated stratospheric footprints in the same polar regions as mentioned for ozone, an upper-tropospheric maximum above the eastern coast of North America and a noticeable minimum on the east from Central America. In the UT, the extratropical $NO_y$ tend to be overestimated, except the hot spot above the eastern coast of North America where $NO_y$ are underestimated. As for ozone, the $H_2O$ meridian variability shown in Fig. 4 is similar between the two data sets, and particularly the delimitation of the area impacted by the Asian monsoon. The simulation catches the geographical maxima above the most convective regions (equatorial lands, and the area impacted by the Asian summer monsoon) and the maximum observed above the tropical Atlantic Ocean, as the collocated ozone minimum. This feature is due to the westward extension of the Central-African peak advected by easterlies (Uma et al., 2014, Fig. 3). However, ozone and water vapour biases illustrate either the difficulty in parameterizing detrainment, notably from tropical convective systems (e.g. Folkins et al., 2006), or the phase of water. The latter depends on temperature but also on supersaturation, which is not implemented in the current model version though it might represent an important fraction of the sampled air masses near the tropopause (Petzold et al., 2020). The LS is characterized by drier values in the model simulation, which is discussed later.







**Figure 1.** Ozone mean horizontal distributions on yearly averages from December 1994 until November 2017, for the products IAGOS-DM (left) and INCA-M (middle), and the biases (right) normalized with respect to the mean values between the two products. Each row displays a layer, with the non-separated UTLS at the top and the distinct LS and UT below.





**Figure 2.** Same as Fig. 1 for carbon monoxide, from December 2001 until November 2017.





**Figure 3.** Same as Fig. 1 for reactive nitrogen, from December 1999 until November 2017.





**Figure 4.** Same as Fig. 1 for water vapour.



## 3.2 Northern extra-tropics

In this section, we propose a synthesis of the assessment in the UT, the LS, and the mixed UTLS, followed by a sensitivity test with respect to the emissions from lightning and from biomass burning. As the tropics are sampled exclusively in the troposphere because of the higher tropopause altitude, we focus on the extra-tropics in order to derive metrics that characterize similar areas between the two layers. Figure 5 shows the scatterplots derived from Figs. 1–4 in the northern extra-tropics, with basic linear regression scores. Table 2 presents complementary metrics as the modified normalized mean biased (MNMB) and the fractional gross error (FGE) defined in Eqs. 1 and 2. For further detail, the seasonal scatterplots are shown in Figs. A1–A4, and the seasonal statistics are presented in Table A1. In this section, it is important to note that the values beyond the 1 and 99 percentiles are excluded from the calculations in order to avoid the scores to be influenced by the most extreme outliers. Concerning the water vapour measurements, it has to be noted that the IAGOS-Core sensor was not initially designed for air masses as dry as in the lower stratosphere and tends to have a wet bias for low RHL values. An additional filter was applied to IAGOS-DM as an attempt to make the LS data usable (see section 2.3.2). However, the comparison between the model and the IAGOS-Core $H_2O$ data in the LS (and in the mixed UTLS) leads to the assumption that the filter was not sufficient, though the latter has been tested down to 5 % without visible change on the MNMB or on the correlation. So, the IAGOS-Core $H_2O$ data cannot be used for model assessment but at the most can be interpreted as upper limit.

### 3.2.1 Model evaluation

According to Table 2, in the mixed UTLS, the core simulation exhibits high geographical correlations for ozone (r=0.96), and relatively high correlations for CO and $NO_y$ (r=0.80 and 0.77 respectively). It suggests that the variations in the tropopause altitude are realistically located in the nudged meteorological fields. The biases in the UTLS are rather negative for ozone and almost systematically positive for CO, and show a wide variability for $NO_y$. Table A1 shows that the annual biases in CO in the UTLS are representative of most seasons. Ozone has relatively small biases except in summer, when it is almost systematically negative. The $NO_y$ species are characterized by negative biases in spring and summer, and positive biases in fall and winter.

More details are provided with the UTLS splitting. For a given species, we denote a high correlation in the layer maximizing the mixing ratio (LS for ozone, UT for water vapour and, to a lesser extent, $NO_y$ in the LS). Except for ozone, the scores regarding biases show better results in the layer maximizing the mixing ratios, i.e. water vapour and CO in the UT and, though with an important variability, $NO_y$ in the LS. The negative bias in lower-stratospheric ozone is characterized by a strong and systematic negative bias in summer (MNMB=-0.302; FGE=0.309) though with a good geographical correlation (r=0.86), and a systematic negative bias in temperature (-2.3 K). The latter suggests that the influence from the deeper stratosphere





is underestimated during this season. On the contrary, good scores are visible for ozone during winter and spring (|MNMB| < 0.06; FGE < 0.12; r ≥ 0.90), suggesting that the impact of the Brewer-Dobson circulation on the LS is well represented. The diagnostics made in this study cannot be used for water vapour in the LS or in the UTLS, despite the filter applied to IAGOS-DM for this species. So far, the current tools used in this study only allow us to assess the model humidity in the UT. However, the modeled climatology (MNMB = -0.552) shows moister values than in ERA5 (not shown) which has been

reported to overestimate the specific humidity in the extratropical LS (Krüger et al., 2022).

      Since their magnitudes are close to their respective FGE, the discrepancies mentioned for water vapour in the LS, ozone and CO display the same sign at most locations. The features concerning CO and $NO_y$ are representative of each season, except summertime $NO_y$ which shows a very low correlation. Mostly representative of summer too, the model also shows more difficulties in simulating the $NO_y$ tropospheric features, especially in the 35–45° N band where high values are seen in

the simulation only (Fig. A3). A comparison (not shown) with a climatology of observed lightning flash rates from the LIS–OTD database (Cecil et al., 2014) showed difficulties from the LMDZ-OR-INCA model to reproduce the lightning geographical distribution, with an important underestimation above marine grid cells and an overestimation above lands. These discrepancies are likely to play a significant role in the poor scores in the modelled $NO_y$ climatologies, and especially during summer when the lightning activity is maximized (e.g. Holle et al., 2016). Uncertainties in aircraft emissions are also a potential source of

important biases for this family of species in the LS, as the LMDZ-OR-INCA model response in $NO_y$ to the aviation emissions can reach more than 450 ppt in every season.

      We note important biases in CO, systematically positive in the LS (MNMB = FGE = 0.232) with a poleward gradient well visible in Fig. 2, and low but negative at most locations in the UT (MNMB = -0.069; FGE = 0.082). As for lower-stratospheric ozone (MNMB = -0.086; FGE = 0.111), the sign of the biases is constant on almost all the sampled locations. Conversely for

water vapour, the represented fraction of the UT is characterized by a positive bias more mitigated geographically (MNMB = 0.071; FGE = 0.142). Complementary information is provided in Table A1 with temperature scores well in phase with the water vapour discrepancies, i.e. a positive bias in the UT with a high geographical variability, and an important correlation in the UT. As for water vapour, this description of the temperature behaviour is representative of most seasons. The saturating vapour pressure and the vertical stability as represented in the model might thus be an important factor in the water vapour

discrepancies. However, the scores do not show the same seasonality between the two variables. The fact that supersaturation is not taken into account in the simulation is one possible reason for this behavioral difference.

      In Fig. 5, we particularly note that the high correlations for ozone in both the UTLS (r=0.96) and the LS (r=0.89), and for water vapour in the UT (r=0.92), are characterized by a linear regression slope close to 1, thus showing a realistic geographical



**Table 2.** Annual metrics synthesizing the assessment of the $O_3$, CO, $NO_y$ and $H_2O$ climatologies from the INCA-M core simulation against IAGOS-DM in several layers, as shown in Fig. 5. From left to right: Pearson's correlation coefficient (r), the modified normalized mean bias (MNMB), the fractional gross error (FGE) and the sample size ($N_{cells}$). As they cannot be used for the model assessment, the results for water vapour in the LS and in the mixed UTLS are represented in brackets. For the temperature, the absolute bias and its associated error are equivalent to the MNMB and the FGE without the normalizing factors.

| Species | Layer | r | MNMB | FGE | $N_{cells}$ |
|---------|-------|------|----------|----------|--------|
| $O_3$ | UTLS | 0.96 | -0.061 | 0.093 | 3,424 |
| | LS | 0.89 | -0.086 | 0.111 | 2,748 |
| | UT | 0.67 | -0.048 | 0.064 | 1,732 |
| CO | UTLS | 0.80 | 0.110 | 0.121 | 3,484 |
| | LS | 0.69 | 0.232 | 0.232 | 2,803 |
| | UT | 0.65 | -0.069 | 0.082 | 1,522 |
| $NO_y$ | UTLS | 0.77 | 0.024 | 0.176 | 3,382 |
| | LS | 0.65 | 0.023 | 0.160 | 2,895 |
| | UT | 0.50 | 0.109 | 0.300 | 1,668 |
| $H_2O$ | UTLS | (0.95) | (-0.158) | (0.193) | 3,346 |
| | LS | (0.73) | (-0.552) | (0.552) | 2,651 |
| | UT | 0.92 | 0.071 | 0.142 | 1,907 |
| | | | Abs. bias (K) | Abs. error (K) | |
| T | UTLS | 0.94 | -0.9 | 1.1 | 3,810 |
| | LS | 0.84 | -1.7 | 1.8 | 3,138 |
| | UT | 0.95 | 0.3 | 1.1 | 2,051 |

variability in these cases. Notably, the meridian structure highlighted with the colors is also well reproduced, and the LMDZ

GCM captures well the large distribution of the water vapour mixing ratio in the low latitudes (orange and red dots), spreading between dry subsiding and wet convective regions. These features concerning water vapour are representative of each season. On the contrary, the lower-stratospheric ozone variability is underestimated in summer and fall. The great scores shown in spring are consistent with a well-reproduced mean impact of the Brewer–Dobson circulation on the ozone mixing ratios, both in spatial distribution and in geographically averaged magnitude. In the UT however, the colors show that the mean ozone

northward gradient is overestimated. Carbon monoxide and reactive nitrogen show lesser scores, with a lower correlation and a more underestimated geographical variability. Concerning $NO_y$, the model reproduces relatively well the lower-stratospheric poleward gradient, probably due to the important quantities of stratospheric nitric acid, but hardly represents the variability inside each latitude band.

### 3.2.2 Comparison with the perturbation runs

The Taylor diagrams presented in Fig. 6 gather a synthesis of the comparison between the reference run and the sensitivity runs, comprising a run without lightning emission ("No-LNOx") and a run without biomass burning emissions ("No-BB"). The aim consists both of understanding further the differences between the reference simulation and the observations, and understanding



further the observed climatologies when the reference run is consistent. In order to represent more clearly the differences between the runs, we chose to draw the mean ratio (with its inter-quartile interval) of the model outputs to the observations.

The advantage is to keep a constant denominator in the normalized mean values, between the different simulations. Since modelled water vapour remains quasi unchanged in the test, only the reference simulation is presented regarding this variable. First, the comparison between the different runs shows a better correlation in the reference simulation in the UT, possibly suggesting that the effects from biomass burning and lightning emissions on ozone production are realistically distributed in space. As expected, no change is observed in the LS for this metric, since the higher amounts of ozone in the LS increase the

$NO_x$ threshold necessary to trigger a net ozone production (e.g. Hegglin et al., 2006). Surprisingly, no important change in the correlation coefficients is obtained for $NO_y$. It is consistent with the fact that areas where lightning emissions are the most abundant also maximize the convective uplift of surface pollutants into the UT. Also, the maximum above the Northeastern American coast is consistent with the higher frequency in warm conveyor belts shown in Madonna et al. (2014). In contrast to $NO_y$, the ozone correlation is sensitive to the removal of lightning sources (r=0.67 compared to r=0.53), suggesting that a part

of the ozone distribution can be explained by the lightning distribution as represented in the model. Concerning CO, we can note a small loss of correlation in the UT without biomass burning or lightnings, but a small increase in the LS as well. While the loss of correlation is consistent for the UT, the gain in the LS may reveal an overestimated tropospheric influence on this layer, as too much convection.

The changes in biases are generally more important with the run without $LNO_x$ than without biomass burning. It diminishes

ozone to an important negative bias (from -15 to -20 % throughout the layers, in annual means), $NO_y$ to a small bias (between -10 and 0 %) and increases CO to a 10–50 % positive bias. The model thus overestimates the non-lightning $NO_y$, but not necessarily the $NO_x$, as ozone is well-underestimated in this simulation, assuming that the shorter and lesser $NO_y$ sampling does not lead to strong differences. There are several possible explanations, including a lack of nitric acid ($HNO_3$) loss by scavenging and/or heterogeneous reactions, and a $HNO_3$ overestimation in the LS due to nitrous oxide degradation in the

stratosphere.

As expected, the impact of biomass burning emissions on the biases is weak for ozone and reactive nitrogen, whatever the season. It decreases the CO mixing ratios such as the annual bias in the UT changes from -5 % to -15 %, from 30 to 15 % in the LS and from 15 to 0 % in the UTLS. Surprisingly, it is not negligible in the LS, especially in summer. It is likely that the influence of biomass burning on the LS is overestimated because of an excessive exchange between the troposphere and the

stratosphere. The change in correlation linked to biomass burning emissions is mainly visible in the upper-tropospheric CO, and is mainly representative of summer, when the r coefficient drops from 0.70 to 0.50. It suggests that this season maximizes





**Figure 5.** Scatterplots representing the INCA-M yearly horizontal climatologies against the IAGOS-DM product, in the latitudes beyond 25°
N. Each row displays a layer, and each column displays a measured variable. Each color represents a latitude band. For each graphic, the
solid black line represents the linear regression fit described on the top-left corner with its equation, its Pearson correlation coefficient and
the amount of grid points involved in its calculation. The grey dashed line illustrates the y=x reference line, surrounded by a shaded +/- 20
% margin. The outliers (outside the 1 and 99 percentiles) are not represented.

the impact of biomass burning in the UT as it contributes significantly to the CO distribution, and it is consistent with the

important summertime maxima in CO emissions from boreal forests in both the GFAS and GFED inventories (Andela et al.,

2013).



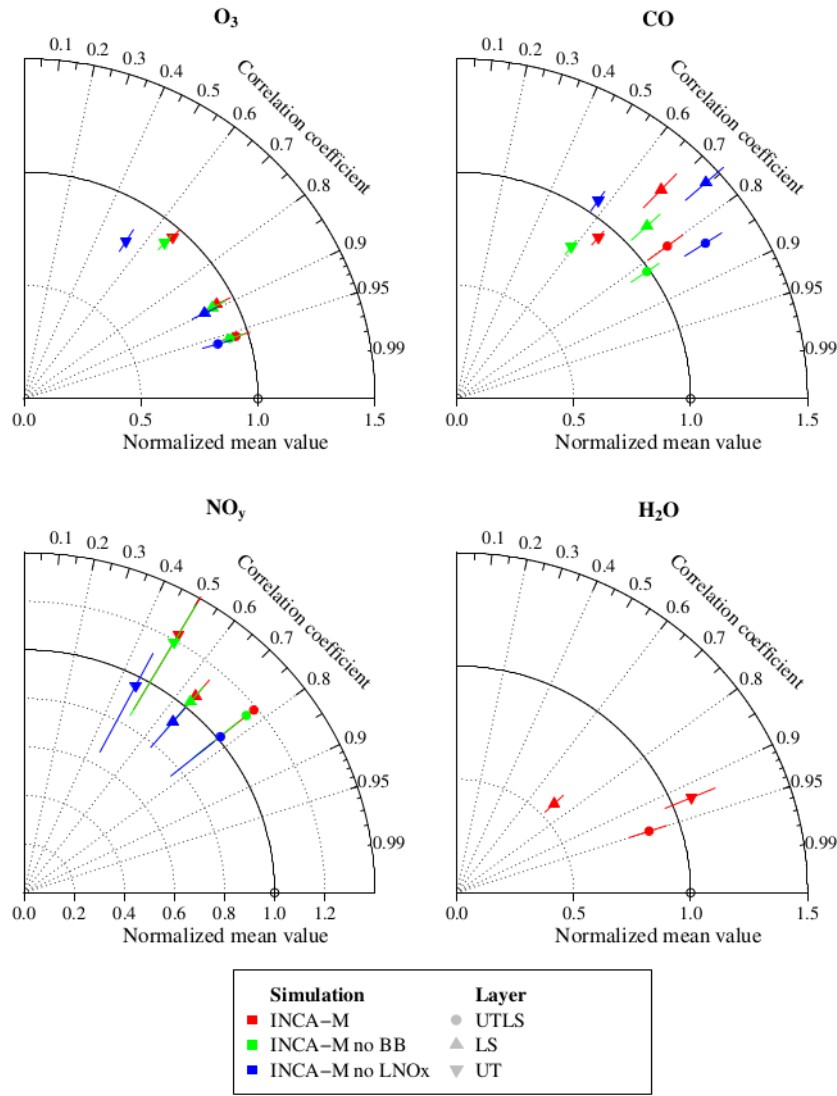

**Figure 6.** Modified Taylor diagrams synthesizing the assessment the yearly climatologies beyond 25° N derived from the three LMDZ-OR-INCA simulations against IAGOS-DM, for $O_3$, CO, $NO_y$ and $H_2O$. Each simulation is represented by a color, and each layer by a point shape. The radial axis corresponds to a normalized mean value. The orthoradial axis refers to the r correlation coefficient. The error bars are the quartiles 1 and 3 of the relative bias.



## 3.3 Tropics


Figures 7–10 compare the zonal cross sections in the tropics derived from IAGOS-DM and the three INCA-M simulations, during the four seasons defined in Table 1. The profiles were derived from averages along both the vertical and longitudinal axes, using the upper-tropospheric grid cells only. The mean pressures on the right axis have been added in order to locate the measurements barycentre and thus to identify changes in mean altitude measurements. They can be associated to significant

changes at the edges of the sampled region, or to change in the width of the longitude interval. This case mainly corresponds to $NO_y$ measurements, during November above Southern Africa and October-November above South Asia. The corresponding profile shapes are thus hardly interpreted, but the comparison with the model remains valuable. Given the negligible changes in water vapour from one simulation to another, we only show its reference simulation profiles, as in Fig. 6. Last, with a lessened sampling efficiency and a shorter measurement time period for $NO_y$, the comparison between its profiles and the ozone profiles

is not necessarily relevant. We thus made a representativeness test on ozone, projecting only the IAGOS data characterized by a valid $NO_y$ measurement. The points where the subsequent difference with the reference ozone profiles is greater than 10 % are indicated with shaded areas in the $NO_y$ panels. Their few occurrences indicate that seasonal mean ozone does not vary much between the two periods and/or sampling modes, which provides more confidence on the representativeness from the $NO_y$ measurements of the whole ozone measurement period.

### 3.3.1 Observed features


Before assessing the model, it is worth presenting the main features exhibited by the observations and proposing some explanation, with a focus on the most complete profiles (Atlantic and Africa). The water vapour maxima are collocated with ozone minima during the northern monsoon seasons (JA and JJASO for the Atlantic and Africa, respectively), representing the most convective areas. Above Africa, in both southern and northern monsoons, Sauvage et al. (2007b) and Lannuque et al. (2021)

attributed the ozone gradients surrounding the minimum to the uplift of precursors in the ITCZ leading to an increased photo-chemical activity during the poleward transport. It is consistent with the peak in the modelled net ozone production efficiency calculated (not shown) that surrounds these ozone minima. In the same continent, the CO maximum is shifted from the water vapour peak. The same study showed that the CO emitted at the surface, notably from the dry areas where biomass burning activity is increased, was uplifted into the ITCZ, transported poleward in the Hadley cell upper branch and accumulating in the

vicinity of increased wind shear areas. Above Atlantic–South America, CO is maximized during SON. Livesey et al. (2013) showed similar results using MLS measurements around 215 hPa from 2004 until 2011, with more significant seasonal cycles above South American tropics and subtropics. They also show this corresponds to the transition season between the continental

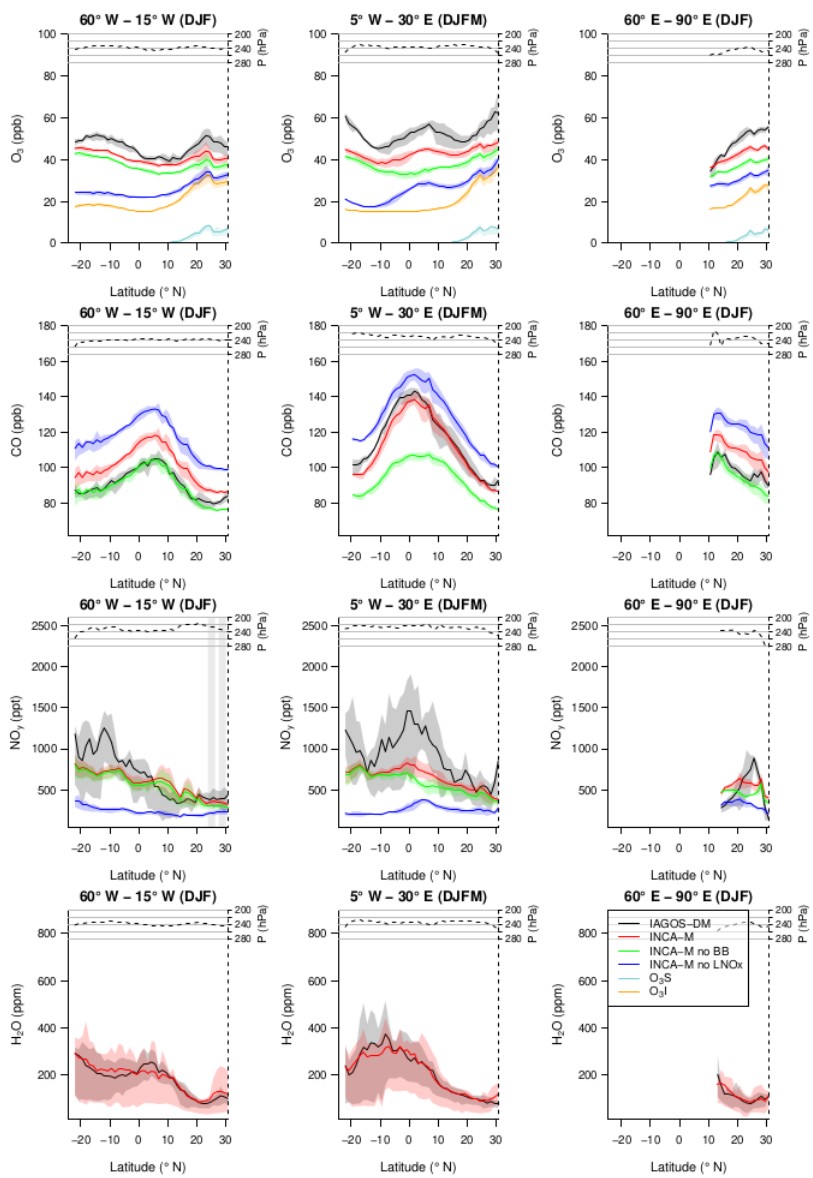

**Figure 7.** Zonal cross sections between 25° S and 30° N from December until February or March. Each row represents a measured variable, and each column represents a longitude interval from which the zonal means have been derived. As the season's definition, they are indicated in the title of each graphic. The uncertainties shown here correspond to the spatial variability, defined as the interval between the quartiles 1 and 3. The solid black line corresponds to IAGOS-DM, whereas the red, blue and green lines correspond respectively to the INCA-M reference simulation, and to the INCA-M simulations without emissions from lightning and from biomass burning. In the ozone panels, the orange and light-blue lines show the $O_3I$ and $O_3S$ stratospheric tracers. The dashed line at the top of each graphic shows the mean pressure derived from observations. The latter's values are reported on the right axis.



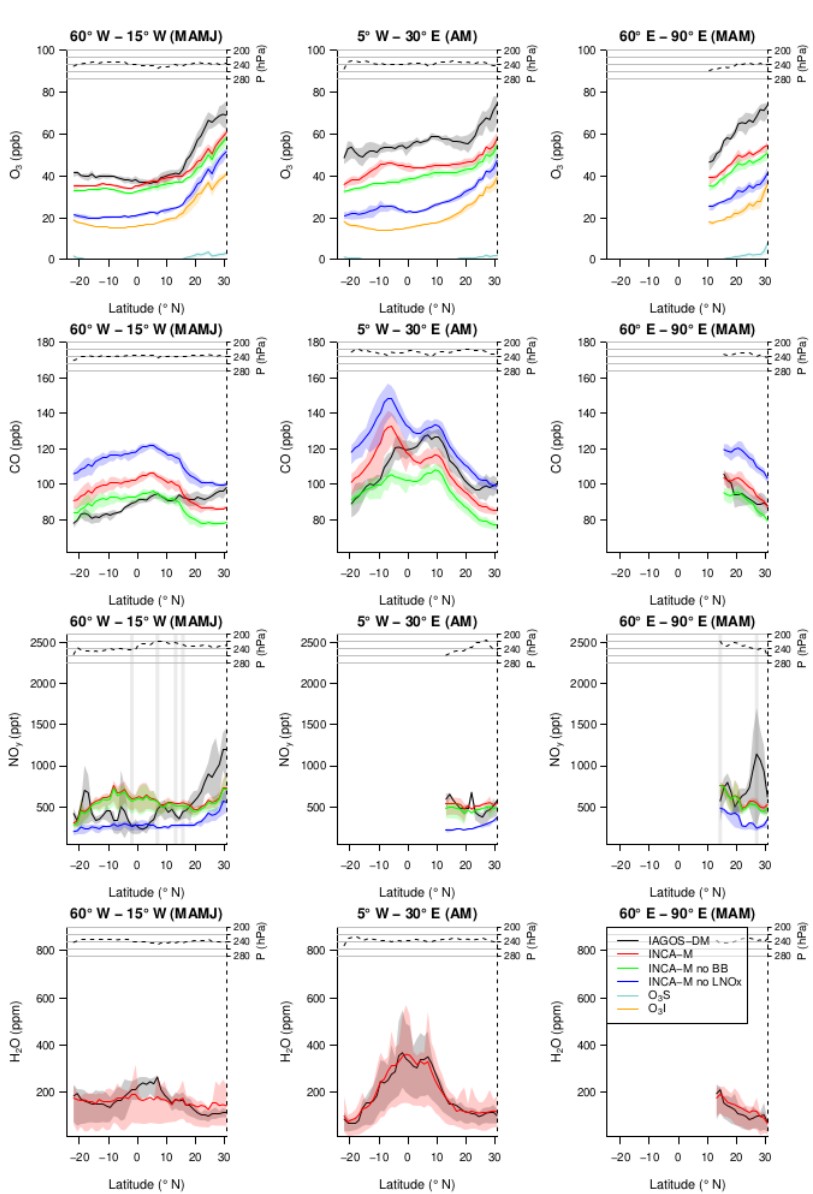

**Figure 8.** Same as Fig. 7 from March or April until May or June.



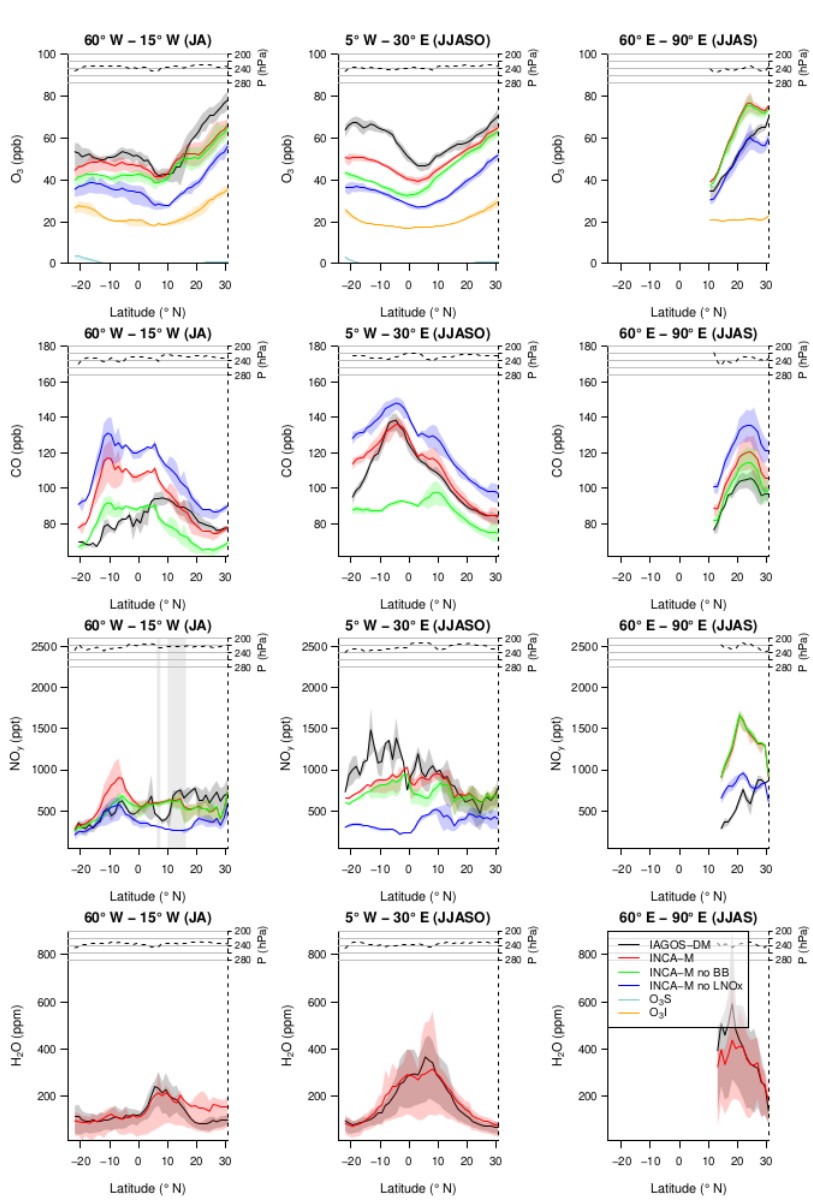

**Figure 9.** Same as Fig. 7 for July–August, June–October and June–September, from left to right.



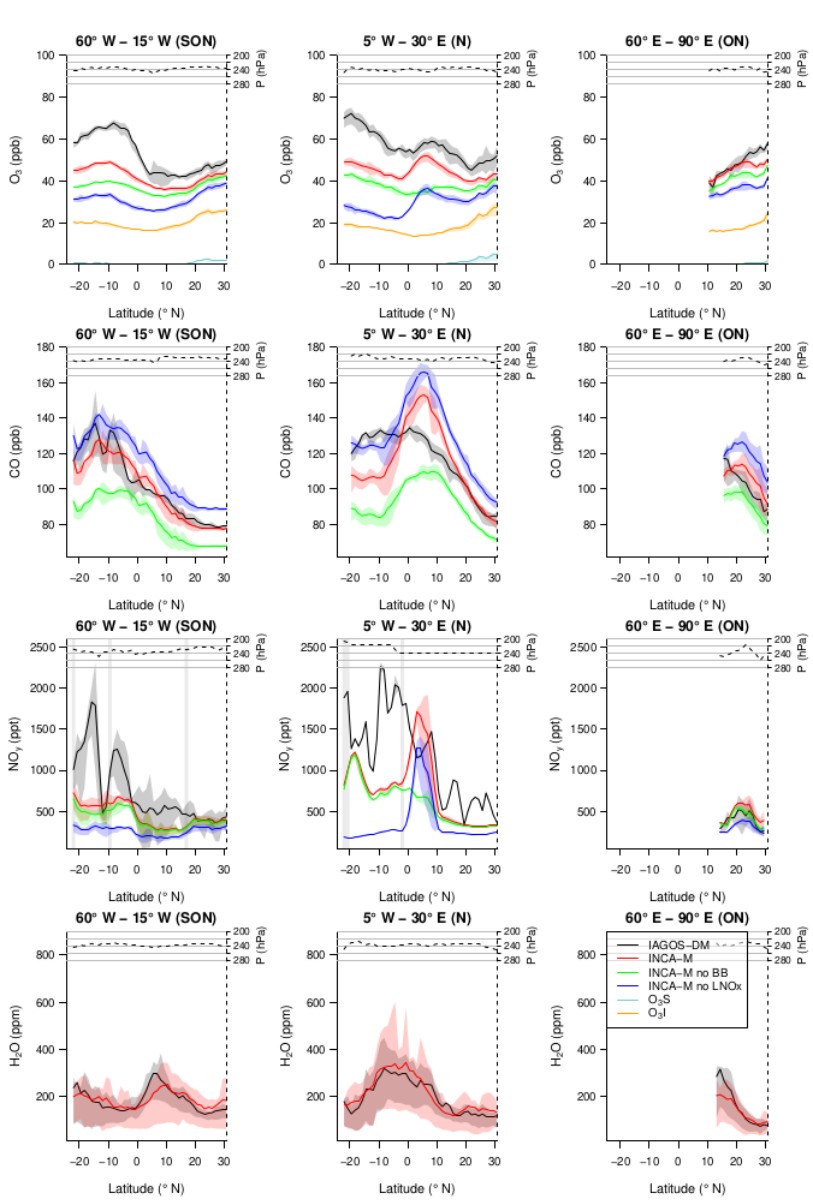

**Figure 10.** Same as Fig. 7 for September–November, November and October–November, from left to right.



dry and wet seasons. The southern CO maximum that we observe here is thus due to the association between the start of an enhanced convective activity while biomass burning emissions are still intense. Among the three regions, tropical Africa shows the most important CO maxima. The only season with comparable peaks between Africa and South America is September–November, and the southern part from 15° S is not likely to be influenced by African emissions, as Yamasoe et al. (2015) showed that these latitudes were characterized by westerly winds during this season. The Asian summer monsoon maximizes the water vapour mixing ratios, reaching 600 ppm against almost 400 ppm above Africa and 300 ppm above South America. This regional maximum may be explained by higher temperatures ($\sim$ +5 K) that allow a more abundant gaseous phase (not shown), and probably due to the particularly strong wet convection. One could expect the CO mixing ratio to be more important in the UT above the Asian summer monsoon, as shown from the Infrared Atmospheric Sounder Interferometer (IASI) satellite data in Barret et al. (2016), with surface tracers accumulating in the associated anticyclone. However, the altitude range observed in Barret et al. (2016) where CO is more abundant in the Asian summer monsoon spreads from 270 up to 110 hPa, thus partially higher than the IAGOS cruise data. It is therefore likely that the higher tropopause altitude characterizing the Asian summer monsoon system (e.g. Fig. 9d in Li et al., 2017) leads to an elevated CO vertical maximum that the IAGOS aircraft cannot sample, as Park et al. (2009) showed a vertical maximum near 15 km inside the anticyclone. In this region, ozone and reactive nitrogen reach their seasonal maxima during March–May, correlated with the lower-stratospheric ozone maximum in the mid-latitudes due to the Brewer–Dobson circulation. It is consistent with the enhanced ozone stratosphere-to-troposphere transport during the pre-monsoon season, well shown in Barret et al. (2016), suggested with the important seasonal $O_3$/CO ratio highlighted in Cohen et al. (2018), and confirmed with measurements from the High Altitude and Long-range Aircraft (HALO) during the HALO-ESMVal campaign in 2012 (Gottschaldt et al., 2018) showing correlated enhancements of hydrochloric acid (HCl) and ozone. This seasonal maximum is then interrupted by the northward shift of the subtropical jet during the monsoon that confines the stratospheric intrusions at the northern side of the Himalayas (Cristofanelli et al., 2010). The strong northward ozone gradient in the monsoon season is consistent with the northward transport of air masses with a seasonally maximized net ozone production (not shown), as simulated in Gottschaldt et al. (2018) also. They linked the important photochemical activity with the combination between uplifted precursors from the surface and lightning $NO_x$ emissions, though the latter were shown reaching their maximum during the previous season.

### 3.3.2 Model assessment

Good consistencies between the reference simulation and the observations are visible for ozone, CO and water vapour. The latter is the species with the best consistency, with the smallest bias at most latitudes and during most seasons. Above the





Atlantic, during the North American summer monsoon (Fig. 9), the model reproduces well the maximum at 5–10° N but not

the drop at the northern side, leading to strong relative biases all along the northern tropic (75 ppm on average, thus 65 % of the

observed mixing ratio). We also note that the model tends to underestimate the latitudinal variability in this region, especially

from March until June (Fig. 8) when it is quasi absent in the simulation. Above Africa, the model captures well the width and

the magnitude of the maximum. Above South Asia, the simulation has difficulties to reproduce the extremely high water vapour

mixing ratios during the monsoon season on average (-110 ppm bias, thus -20 %). Still, water vapour remains simulated with

higher amounts in the UT above the Asian summer monsoon than above the other regions. Despite these significant biases, the

overall consistency in water vapour profiles suggests that the transport in the nudged simulation is reliable and can reproduce

accurately some convective features, even in the monsoon systems.

Ozone is almost systematically underestimated in the reference simulation but its variations are mostly in agreement with the

observations, with collocated extrema and similar meridional gradients. The stratospheric ozone tracer ($O_3S$) indicates very low

values, systematically less than 5 ppb except during the DJF/DJFM season when it plays the main role in the northward ozone

gradient north of 15–20° N. However, we note an underestimated northward gradient in the northern subtropics, especially

during the March–May season. Though this season maximizes the stratosphere-to-troposphere transport as explained in the

previous paragraphs, the $O_3S$ tracer shows low mixing ratios, which highlights an underestimated impact from the stratospheric

intrusions. The inert stratospheric ozone tracer ($O_3I$), instead, follows a stronger gradient in this area. The underestimation of

the stratospheric influence in INCA-M may thus be explained by an underestimation of the ozone lifetime in these areas and

seasons. Carbon monoxide tends to be overestimated, except above Africa since December until March and since June until

October when the profiles are particularly well reproduced, combining good correlations and small biases. In most regions and

seasons, the simulation shows a consistent variability in CO despite some cases where the profiles are poorly correlated with

the observations (mainly the MAMJ and JA seasons over the Atlantic Ocean). The model reproduces well the higher maximum

CO mixing ratios in tropical Africa, compared to the other two areas. The $NO_y$ profiles underestimate the meridian variability.

Above Africa, they are almost systematically underestimated in the southern hemisphere, but they generally show consistent

values in the northern hemisphere. Last, we note an important positive bias during the Asian summer monsoon (more than

+100 % on average) that is further characterized later, using the other two simulations.

### 3.3.3   Comparison with the perturbation runs

As expected, the lightning emissions have a stronger contribution to upper-tropospheric ozone compared to biomass burning,

as suggested by a similar behaviour for $NO_y$. Though the source strengths are comparable, the important contribution from



lightning to the $NO_x$ injection at these altitudes leads to a greater ozone production efficiency, compared to other sources

(Sauvage et al., 2007a). Notably, the Atmospheric Chemistry and Climate Model Intercomparison Project (ACCMIP) models

estimated the ozone production efficiency from lightning to be $6.5 \pm 4.7$ times greater than from the other sources (Finney

et al., 2016). Lightning emissions also contribute significantly to the meridional gradients in ozone and $NO_y$ north and south

of the ITCZ, as the difference between the reference and the no-$LNO_x$ simulations shows an important variability. As expected

also, the role of lightning $NO_x$ into CO destruction mostly consists of a background signal, involving $NO_x$ emissions that

enhance both ozone and OH production, ozone itself acting as a source of OH in presence of water vapour. The increased OH

mixing ratios finally destroy CO with an average lifetime of 38 days in the tropics (Lelieveld et al., 2016). The CO chemical

destruction is thus a slow process compared to zonal transport, which can explain the spread pattern of the sensitivity to $LNO_x$

emissions. Some geographical differences in the impacts of lightning on CO are still visible, notably between two opposite

subtropics, the latter probably reflecting a slow interhemispheric transport.

Some ozone discrepancies can be explained by the combined comparison between species and between simulations. For

example, the ozone and CO local maximum simulated near $5$–$10°$ S over Africa in April–May is not visible in the observations.

This increase remains visible in the no-$LNO_x$ simulation but not in the no-BB simulation. It is particularly visible in the CO

profiles, characterized by an exaggerated peak collocated with the ozone local maximum. The impact of biomass burning is

therefore overestimated in the model over this area during April–May. A similar feature is highlighted in November above

Africa, where a peak in $NO_y$ is seen only by the model and arises from biomass burning. This overestimation in biomass

burning products contributes to a collocated steep peak in CO whereas the observations show a flat maximum, and to an ozone

local maximum while it is barely visible in the observations. Since even the no-BB simulation exhibits a peak in CO that

contrasts with the IAGOS-DM flat maximum, the convection parameterization and/or the anthropogenic emission inventory

may play a role in this overestimated spatial variability. Last, one noticeable ozone discrepancy takes place during the Asian

summer monsoon, when the bias reaches +20 ppb. The $NO_y$ profiles allow us to point out the excessively high modelled

value, reaching more than twice the observed mixing ratios. It is interesting to note that even without $LNO_x$, $NO_y$ remain

overestimated and ozone becomes more consistent with the observed profile. Since the impact of lightning activity during this

monsoon on ozone production is well established (e.g. Gottschaldt et al., 2018), it suggests either an overestimated transport

from the boundary layer, or an underestimated washout of soluble species like $HNO_3$.

These sensitivity tests also allow us to associate significant contributions to several well-reproduced features. Above South

America–Atlantic Ocean, the CO maximum during SON between 5 and $15°$ S has a non negligible contribution from local

biomass burning ($\sim 20$ ppb, thus $\sim 10$ ppb more than in other latitudes), consistently with the literature (notably Livesey et al.,



2013; Tsivlidou et al., 2022). The lightning contribution to the ozone maximum between 5 and 15° S is in agreement with the GEOS-Chem model used in Yamasoe et al. (2015). The next season (DJF) is characterized by a well correlated CO profile

though positively biased, and the model associates the 5° S–15° N maximum to other sources. During the summer monsoon above Africa, the CO peak above 0–10° S is associated to local biomass burning emissions, as a significant part of the peak above 5° S–5° N during the opposite season (DJFM). On the contrary, the observed maximum during April–May between 5 and 10° N is rather associated to other sources. These features are in agreement with the results presented in Lannuque et al. (2021) based on the SOFT-IO source-apportionment software (Sauvage et al., 2017). According to the model, an important part

of the differences in CO between tropical Africa and the other two regions are mainly due to biomass burning. Above South Asia, CO is less influenced by biomass burning during the monsoon season, consistently with the literature. For example, Jiang et al. (2007) attributed most of upper-tropospheric CO levels to anthropogenic emissions, because of deep convection that both uplifts surface pollution into the UT and reduces wildfires with enhanced precipitations.

## 4 Summary and conclusions

This study consists of the assessment of a long-term simulation from the LMDZ-OR-INCA chemistry-climate model (CCM) with daily resolved outputs in the upper troposphere–lower stratosphere (UTLS). More precisely, we evaluate ozone, carbon monoxide (CO), reactive nitrogen ($NO_y$) and water vapour climatologies based on all the cruise IAGOS data set including the IAGOS-CARIBIC data, respectively during the periods Dec. 1994–Nov. 2017, Dec. 2001–Nov. 2017, Dec. 1999–Nov. 2017 and Dec. 1994–Nov. 2017.

In order to allow a direct comparison between the simulation output and the high-resolution IAGOS data sets, we use the Interpol-IAGOS software that projects the IAGOS data onto the model grid (Cohen et al., 2021). As a first step, we extend this tool to daily model outputs. The subsequent IAGOS product (IAGOS-DM) is generated by interpolating the IAGOS data onto the model grid, then deriving weighted monthly averages on each grid cell. Similar to IAGOS-DM, the product based on the simulation output (INCA-M) is also made of monthly averages across the sampled daily gridpoints only. As a second step, we

compare the annual and seasonal climatologies derived from these two products. The assessment in the mid-latitudes is made separately in the upper troposphere (UT) and the lower stratosphere (LS) using the model potential vorticity (PV), but also in the UTLS like in a single layer, as an option for the models that do not sort out the potential vorticity. In the tropics, the assessment only accounts for upper-tropospheric air masses because of the higher tropopause altitude.

In the northern mid-latitudes, the LMDZ-OR-INCA model exhibits good skills for ozone in the LS, and for water vapour

in the UT. The seasonal scores show that the influence from the deeper stratosphere on the LS through the Brewer–Dobson





circulation is well modelled. At most locations, ozone is slightly underestimated in the UT, and CO shows a positive bias in the LS and a slight negative bias in the UT. These features suggest an overestimation in the extra-tropical cross-tropopause mixing. The bias in reactive nitrogen shows an important geographical variability in every layer. It is likely linked with the difficulty in reproducing the lightning geographical distribution, but also with aircraft emissions, as shown by some biases in the shape of tracks. The latter can play a significant role in $NO_y$ levels. For example, the model intercomparison presented in Olsen et al. (2013) shows an aviation $NO_y$ perturbation ranging from 15 to 40 % of the $NO_y$ level at the cruise altitudes, suggesting an important sensitivity to aircraft emissions. Last, another possible cause for the $NO_y$ discrepancies is the uncertainty in the scavenging processes for soluble species like $HNO_3$ during their upward transport. The IAGOS-Core humidity sensor was initially designed for tropospheric air masses. Though a filter has been applied in an attempt to exclude most of the measurements likely to overestimate the humidity, the corresponding climatologies in the LS shown in this study cannot be used to assess the model simulation.

In the tropics and subtropics, the mean zonal cross sections are generally in good agreement between the model and the observations for ozone, CO and especially for water vapour. The latter shows that the LMDZ model, nudged into the ERA-Interim reanalysis, is able to represent accurately the mean transport features, notably the water vapour geographical maximum in the Asian summer monsoon. CO is well represented in the regions and seasons characterized by important contributions from biomass burning, i.e. during the convective season above South America, and above Africa for the seasons with the southernmost (December–March) and the northernmost (June–October) shifts of the ITCZ. In these cases, the model attributes respectively 25 ppb, 30 ppb and 45 ppb of the CO peaks to biomass burning, and attributes between 10 and 20 ppb of the CO sink to lightning emissions. Though ozone is generally underestimated, the extrema locations and the meridional gradients are consistent with the observations in most seasons and longitude domains. It is mostly sensitive to lightning emissions of nitrogen oxides ($LNO_x$) which can contribute up to a half of the modelled ozone in the southern hemisphere during the first half of the year. On the other hand, the biomass burning contribution to modelled ozone reaches 20–25 % where CO attributed to biomass burning peaks.

Some of the inconsistencies in ozone and CO with respect to the observations are linked to biomass burning emissions. Consequently, improvements in the biomass burning emissions or convection up to the UT is likely to enhance the skills for CO and, to a lesser extent, for ozone. Also, though lightning as represented in the model helps in understanding the ozone geographical distribution, improving the lightning parameterization is likely to lead to enhance the skills for $NO_y$ and ozone.

As demonstrated through this paper, the new version of the Interpol-IAGOS software allows a multi-species assessment for modelled climatologies in the separated UT and LS, or in the UTLS as a whole, by using either the model daily output as well



as the model monthly output (Cohen et al., 2021). It can easily be applied to a wide range of long-term simulations, notably in multimodel experiments. Concerning the latter, two applications are currently in progress in the frame of the second phase of the Tropospheric Ozone Assessment Report (TOAR-II) and of the ACACIA EU project (Advancing the Science for Aviation and Climate), and will be published elsewhere. Another potential application is the assessment of modelled time series on regional scales through seasonal cycles, interannual variability and long-term trends, possibly allowing a source apportionment

in the observed features.



## Appendix A: Seasonal scatterplots in the northern extra-tropics

**Figure A1.** Same as Fig. 5 for boreal winter.





**Figure A2.** Same as Fig. 5 for boreal spring.





**Figure A3.** Same as Fig. 5 for boreal summer.





**Figure A4.** Same as Fig. 5 for boreal fall.



**Table A1.** Same as Table 2 for each season.

| Species | Layer | Season | r | MNMB | FGE | $N_{cells}$ | Season | r | MNMB | FGE | $N_{cells}$ |
|---|---|---|---|---|---|---|---|---|---|---|---|
| $O_3$ | UTLS | DJF | 0.95 | 0.063 | 0.134 | 3,522 | JJA | 0.94 | -0.220 | 0.244 | 3,289 |
| | LS | | 0.88 | 0.053 | 0.116 | 2,965 | | 0.86 | -0.302 | 0.309 | 2,372 |
| | UT | | 0.34 | -0.029 | 0.088 | 1,287 | | 0.44 | -0.030 | 0.077 | 1,921 |
| CO | UTLS | | 0.76 | 0.175 | 0.177 | 3,606 | | 0.79 | 0.084 | 0.114 | 3,255 |
| | LS | | 0.57 | 0.278 | 0.278 | 2,992 | | 0.71 | 0.232 | 0.235 | 2,354 |
| | UT | | 0.59 | -0.019 | 0.059 | 1,092 | | 0.70 | -0.064 | 0.087 | 1,700 |
| $NO_y$ | UTLS | | 0.76 | 0.267 | 0.334 | 3,105 | | 0.37 | -0.102 | 0.279 | 2,702 |
| | LS | | 0.56 | 0.235 | 0.292 | 2,580 | | 0.18 | -0.130 | 0.261 | 1,836 |
| | UT | | 0.22 | 0.344 | 0.519 | 763 | | 0.25 | 0.141 | 0.327 | 1,121 |
| $H_2O$ | UTLS | | (0.93) | (-0.300) | (0.317) | 3,283 | | (0.95) | (-0.087) | (0.174) | 3,240 |
| | LS | | (0.79) | (-0.545) | (0.545) | 2,673 | | (0.65) | (-0.557) | (0.558) | 2,235 |
| | UT | | 0.87 | 0.033 | 0.182 | 1,377 | | 0.93 | 0.181 | 0.230 | 2,118 |
| | | | | Abs. bias (K) | Err. (K) | | | | Abs. bias (K) | Err. (K) | |
| T | UTLS | | 0.96 | -1.1 | 1.3 | 3,802 | | 0.95 | -0.7 | 1.1 | 3,587 |
| | LS | | 0.95 | -1.5 | 1.6 | 3,240 | | 0.85 | -2.3 | 2.4 | 2,674 |
| | UT | | 0.94 | 0.2 | 1.3 | 1,538 | | 0.93 | 1.3 | 1.7 | 2,230 |
| $O_3$ | UTLS | MAM | 0.96 | -0.046 | 0.102 | 3,192 | SON | 0.93 | -0.083 | 0.120 | 3,624 |
| | LS | | 0.90 | -0.030 | 0.099 | 2,745 | | 0.81 | -0.139 | 0.162 | 2,782 |
| | UT | | 0.39 | -0.099 | 0.115 | 1,340 | | 0.61 | -0.031 | 0.071 | 1,802 |
| CO | UTLS | | 0.85 | 0.077 | 0.138 | 3,339 | | 0.73 | 0.143 | 0.147 | 3,574 |
| | LS | | 0.73 | 0.187 | 0.197 | 2,823 | | 0.51 | 0.269 | 0.269 | 2,853 |
| | UT | | 0.58 | -0.163 | 0.169 | 1,138 | | 0.73 | -0.025 | 0.057 | 1,570 |
| $NO_y$ | UTLS | | 0.77 | -0.162 | 0.247 | 2,932 | | 0.68 | 0.236 | 0.298 | 2,861 |
| | LS | | 0.60 | -0.104 | 0.222 | 2,544 | | 0.55 | 0.242 | 0.281 | 2,170 |
| | UT | | 0.36 | -0.199 | 0.419 | 782 | | 0.37 | 0.165 | 0.331 | 1,022 |
| $H_2O$ | UTLS | | (0.93) | (-0.190) | (0.240) | 3,171 | | (0.93) | (-0.221) | (0.252) | 3,410 |
| | LS | | (0.71) | (-0.548) | (0.551) | 2,630 | | (0.69) | (-0.542) | (0.543) | 2,634 |
| | UT | | 0.88 | 0.087 | 0.173 | 1,560 | | 0.89 | 0.042 | 0.151 | 1,902 |
| | | | | Abs. bias (K) | Err. (K) | | | | Abs. bias (K) | Err. (K) | |
| T | UTLS | | 0.91 | -0.6 | 1.0 | 3,618 | | 0.92 | -1.3 | 1.5 | 3,942 |
| | LS | | 0.84 | -1.1 | 1.3 | 3,216 | | 0.79 | -2.1 | 2.2 | 3,143 |
| | UT | | 0.93 | 0.7 | 1.4 | 1,605 | | 0.95 | 0.0 | 1.0 | 2,066 |

**Appendix B: Seasonal Taylor diagrams in the northern extra-tropics**



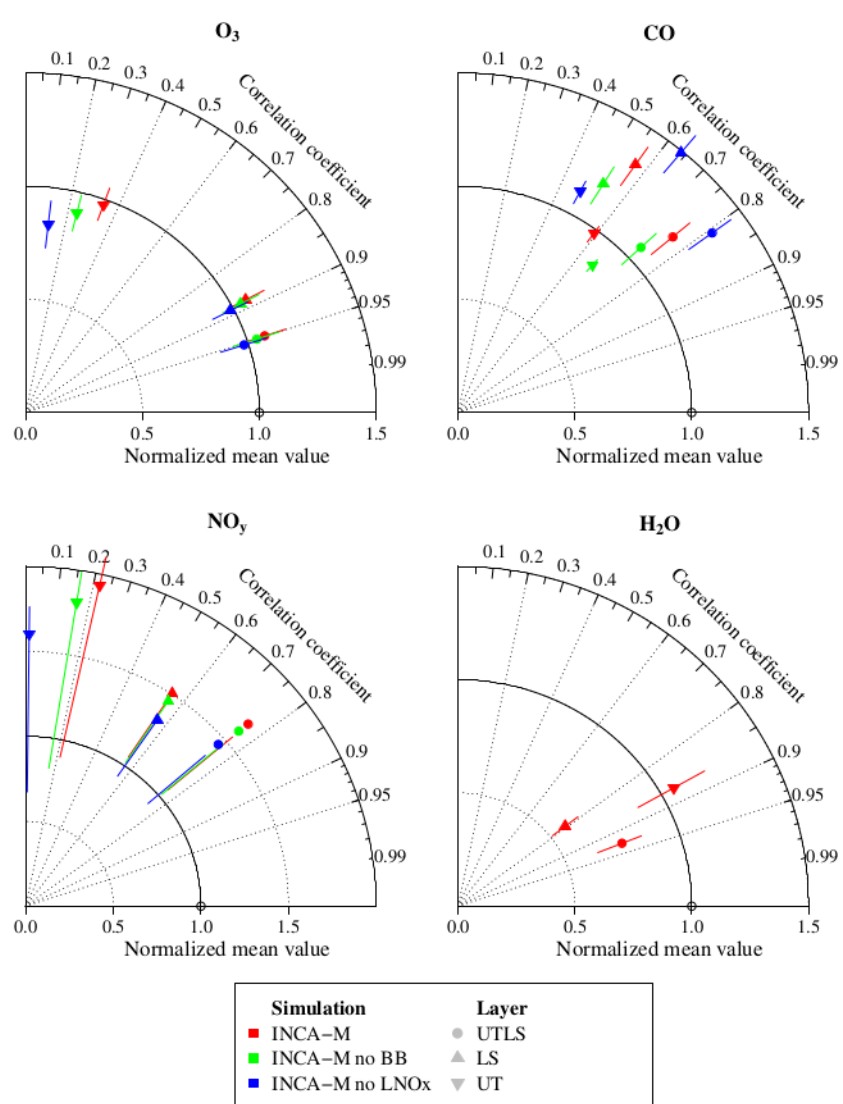

**Figure B1.** As Fig. 6 for boreal winter.





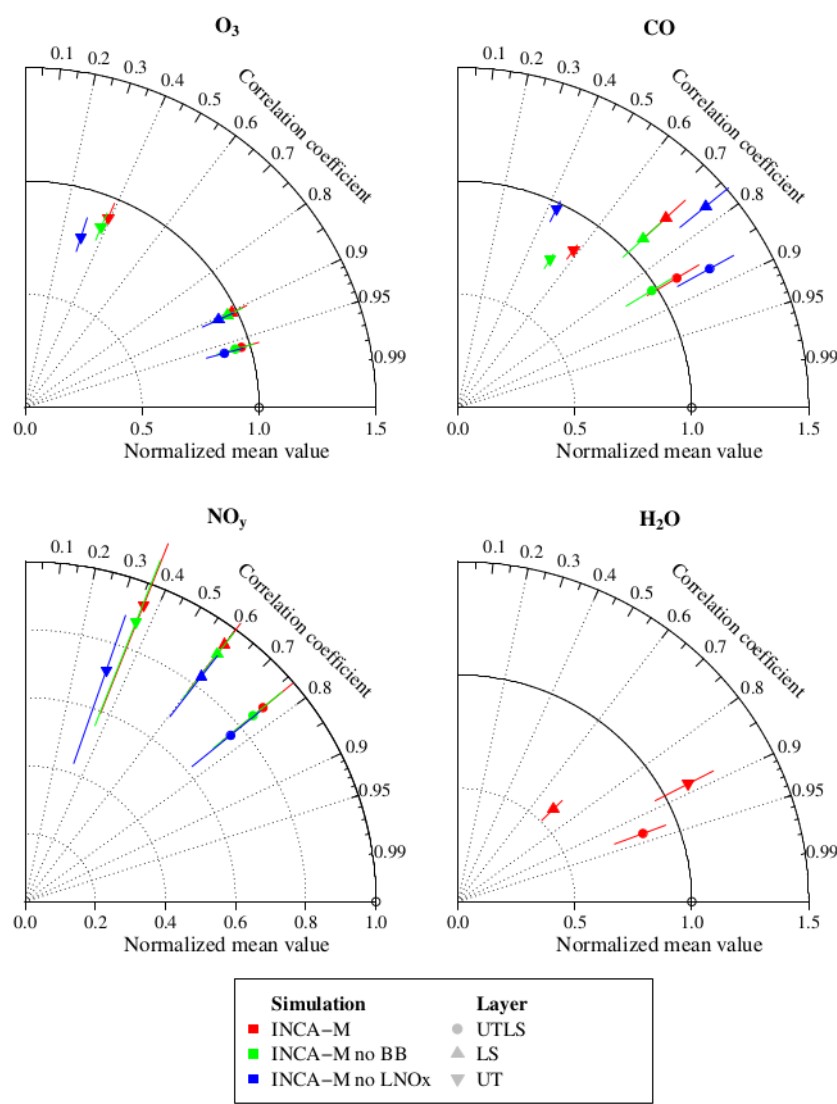

**Figure B2.** As Fig. 6 for boreal spring.





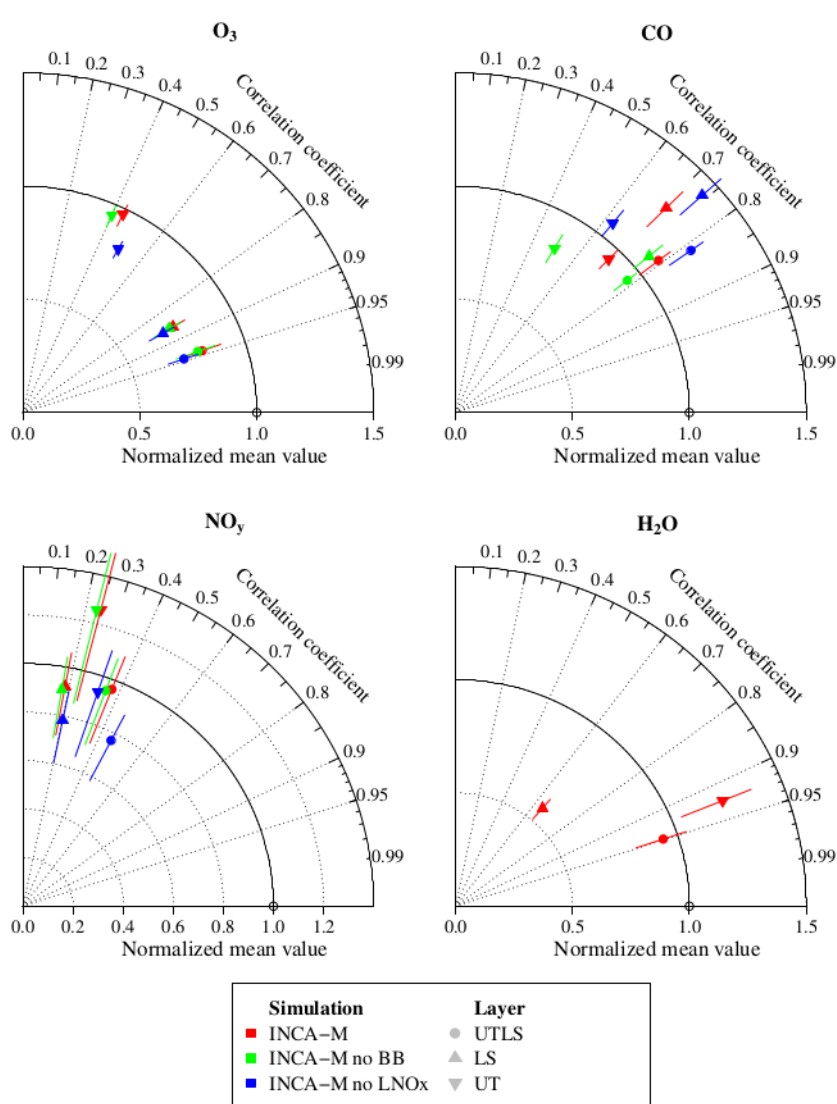

**Figure B3.** As Fig. 6 for boreal summer.



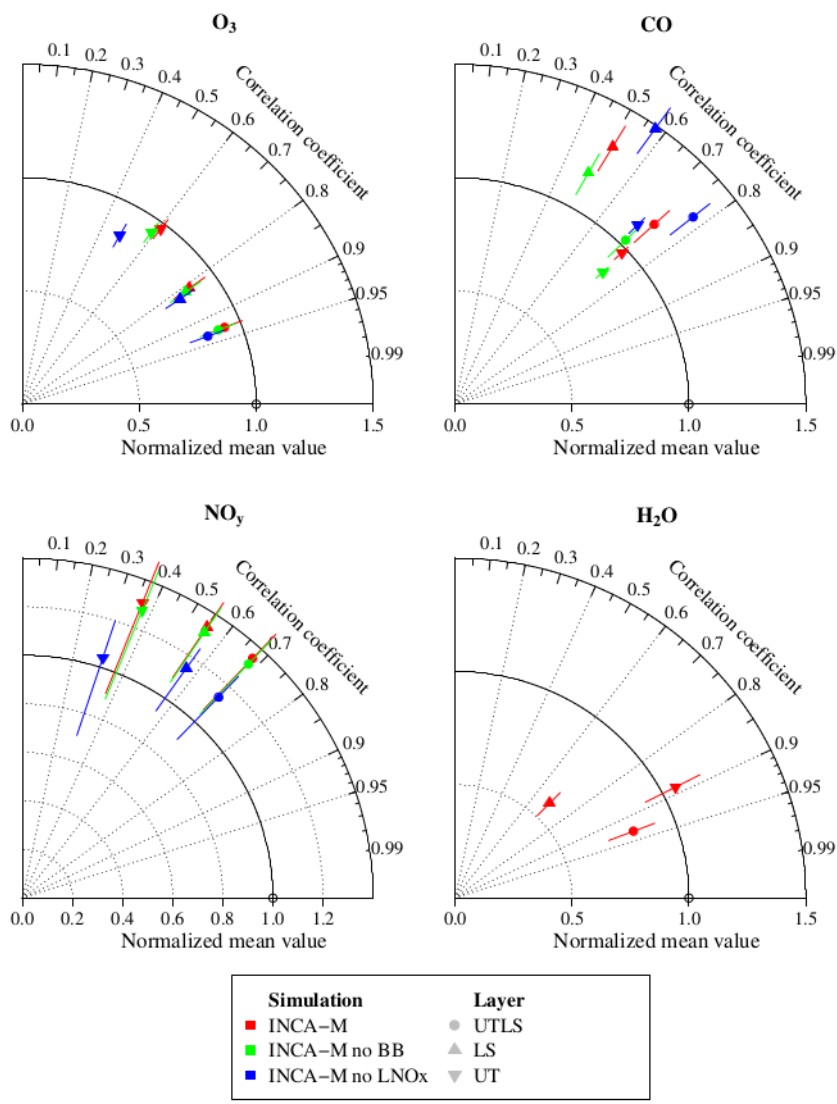

**Figure B4.** As Fig. 6 for boreal fall.

*Code and data availability.* The IAGOS data (IAGOS, 2022) are available at the IAGOS data portal (https://doi.org/10.25326/20) and more precisely, the time series data are found at https://doi.org/10.25326/06 (Boulanger et al., 2018). The Interpol-IAGOS software is available at

https://doi.org/10.25326/81 (Cohen et al., 2020).

*Author contributions.* YC designed the study, with advice from DH and BS. The Interpol-IAGOS software was further developed by YC. The simulations output were provided by YC and DH. The IAGOS data were provided by VT, AP, SR, UB, AZ and HZ. PK and SR



provided important advice on the use of the water vapour data, and DH on the use of the model. The paper was written by YC and reviewed, commented, edited and approved by all the authors.

*Acknowledgements.* The authors acknowledge the strong support of the European Commission, Airbus, and the Airlines (Lufthansa, Air-France, Austrian, Air Namibia, Cathay Pacific, Iberia and China Airlines so far) who carry the IAGOS-Core equipment and perform the maintenance since 1994. In its last 10 years of operation, IAGOS-Core has been funded by INSU–CNRS (France), Météo-France, Université Paul Sabatier (Toulouse, France) and Research Center Jülich (FZJ, Jülich, Germany). IAGOS has been additionally funded by the EU projects IAGOS-DS and IAGOS-ERI. The IAGOS-Core database is supported by AERIS. Data are also available on the AERIS web site

www.aeris-data.fr. The simulations were performed using HPC resources from GENCI (Grand Équipement National de Calcul Intensif) under the gen2201 project. We also wish to acknowledge our colleagues from the IAGOS teams in FZJ, LAERO, DLR and KIT for all the preparation of the IAGOS and CARIBIC data used in this study, and the colleagues in LSCE for the training on the use of the modelling tools.

*Financial support.* This research has been funded by the European Union Horizon 2020 research and innovation programme under the

STRATOFLY (grant agreement no. 769246) and ACACIA (grant agreement no. 875036) projects, and by the Direction Générale de l'Aviation Civile (DGAC) under the ClimAviation project.

*Competing interests.* The authors declare that they have no conflict of interest.



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
