# Peer review of "Evaluation of modelled climatologies of $O_3$ , CO, water vapour and $NO_y$ in the upper troposphere–lower stratosphere using regular in situ observations by passenger aircraft"

_EGUsphere, 2023_

## Author Comment (AC1)

**Authors' response to Reviewer 1**

We are thankful to both reviewers for their positive and accurate feedback on our study, and for the improvements they helped us make to this paper.

The responses are organized as follows: the reviewer's comment is in blue, our answers are in black, and the changes proposed for the revised manuscript are in italic (black for modified sentences, grey for unchanged sentences that have been pasted here in order to remind the context).

The manuscript presents a comparison of O3, CO, NOy and H2O from a nudged simulation of the LMDZ-OR-INCA chemistry-climate model with a set of long-running observations of these trace gases from instrumentation carried on-board commercial aircraft. The comparison is made over the upper troposphere / lower stratosphere region from observations made during cruise-level flights using the Interpol-IAGOS software package described in an earlier paper. To account for the strong discontinuities around the tropopause, separate comparisons can be made for the lower stratosphere and the upper troposphere. In addition, the comparison to observations are made for two sensitivity simulations: one with emissions from biomass burning turned off, and one with emissions from lightning turned off.

The observations shows many of the large scale features of the distribution of these trace gases associated with, for example, the differing height of the tropopause between the tropics and the mid-latitudes, the seasonal and regional nature of biomass burning emissions and the impact of monsoon circulations. The comparison with the model shows the model has an ability to reproduce many of these features, with the most significant and widespread bias being for CO in the lower stratosphere. The paper is generally well organized, though the use of particular phrases or words makes comprehension challenging in a few places. I have tried to point these out in the minor comments.

My only significant concern is the treatment of the vertical height coordinate when comparing the aircraft observations to the model. From Section 2.3.1, I can understand how the IAGOS observations are horizontally gridded. And I can understand how the IAGOS data is binned for the version of the data separated into the upper troposphere and the lower stratosphere. With most (or all?) of the data from cruise altitude, what I am missing is how variations in the cruise altitude are treated. At line 197 it is stated 'The climatologies here refer to nearly horizontal maps derived from partial columns in the cruise altitudes.' Are the point measurements assumed to be representative of a certain vertical range to allow a partial column to be calculated? Or are all the different aircraft observations in a particular month kept on their individual altitude points and these are combined to produce a vertically integrated (or vertically averaged) quantity at each grid point? Or

maybe the relatively small variations in altitude are ignored and all data is assumed to be on an average cruise altitude? It is particularly important for the comparison in the lower stratosphere, which shows such strong vertical gradients in CO and ozone.

We thank Reviewer 1 (R1) for pointing out this need of clarification. The altitude of the measurements is taken into account during the interpolation step, by the use of the weighting coefficients (as for the horizontal axes). Once the 3D climatological means are calculated, we derive the vertical averages from the grid cells that meet the sampling criteria.
Thus, the "partial columns" are not necessarily representative of a geographically constant altitude, but the altitude range remains consistent between the model and the gridded observations.

The first phrase of the subsection 2.3.1 has been completed:
*"The strategy consists of adapting the IAGOS data to the studied simulation in matter of spatial resolution, following a linear reverse interpolation **onto the three spatial dimensions**."*

Also, the term "partial columns" has been replaced by "vertical means" or "vertical averages".

Is it sufficient just to classify observations at 'lower stratosphere' without taking into account the distance to the tropopause?

It depends on the accuracy one wants to give to the assessment of the model. Dividing the lower stratosphere into sub-layers would bring further information on the transport processes between troposphere and stratosphere, and could be tested in the next study. The assumption of the LS as a uniform layer is not realistic geophysically, but it still brings relevant information on the model abilities to reproduce the UTLS behaviour.

Minor comments:

Line 2 – The use of 'the latter' in 'The latter is regularly...' does not work here because the preceding sentence does not present two options.
According to the comments from R1 and R2, the formulation has been changed into:
*"Evaluating global chemistry models in the upper troposphere - lower stratosphere (UTLS) is an important step toward **an improved understanding of the chemical composition in this region. This composition** is regularly sampled [...]"*

Line 15 – is there a word missing in 'as [are] the observed CO peaks due to biomass burning...'
Following the suggestions from R2, the phrase has been reformulated into:

*"In the tropics, the upper-tropospheric climatologies are remarkably well simulated for water vapour; they also show well-reproduced CO peaks due to biomass burning in the most convective systems, and the ozone latitudinal variations are correlated between the observations and the model."*

Line 139 – Does 'In this study, the LMDZ GCM surface zonal and meridional wind components are nudged…' mean that only the surface winds (lowest model layer) are nudged and the rest of the atmosphere is allowed to freely evolve?
There has been a mistake in the manuscript (which has been corrected): the horizontal winds are nudged at every level, not only at the surface.

Line 168 – 'precising' might be better as 'denoting'
This suggestion has been applied.

Line 179 – what is meant by 'if it is not adjacent to the 2 PVU isosurface'? What comes between a particular grid point and the 2PVU isosurface so that it is not adjacent?
This phrase has been clarified as follows:
*"a sampled grid point is considered as upper-tropospheric if its PV is lower than 2 PVU, **if it is not the first gridpoint below the 2 PVU isosurface,** and [...]"*

Lines 173 – 195 – A more general comment on Section 2.3.2: There is no discussion of how PV is used to classify the IAGOS measurements. I assume reanalysis data is interpolated in space and time to each IAGOS measurement and PV is calculated with the same exclusion of the transition layer?
Though there exists a IAGOS product which interpolates the ERA-Interim PV onto the aircraft trajectories, it is not used in this study. In our case, the classification is the same between the model gridpoints and the IAGOS-DM gridpoints, as it only accounts for the model daily PV field.
In order to clarify it, we added the following phrase at the end of the paragraph:
*"It is worth precising that the same classification applies between the INCA-M and the IAGOS-DM grid points, using the model PV field."*

Line 188 – I think 'Consequently, it is worth figuring out that…' would be better as 'Consequently, it is worth keeping in mind that…' or maybe 'worth remembering…'?
The former suggestion has been applied.

Line 219 – I am not sure what is meant by 'In the tropics, the threshold is adapted to the seasons duration by applying a cross product.' A cross product is usually an operation on vectors in linear algebra.
The formulation has been corrected, as follows:
*"In the tropics, the threshold is adapted **proportionally to the seasons duration**."*

Line 349 – For 'we chose to draw the mean ratio', in place of 'draw' that can have

different meanings, could I suggest 'plot' or 'display'?
Agreeing with R1, we replaced the verb "draw" by "display".

Lines 353 – 354 – I think it is a bit of a jump to suggest that biomass burning and lighting are realistically distributed because the simulations without these important sources has a poorer comparison with observations.
This phrase has been replaced by:
*"First, the comparison between the different runs shows a better correlation in the reference simulation in the UT,* **implying that the impacts from lightning and biomass burning in the reference simulation contribute to a non-negligible part of the geographical similarities between IAGOS-DM and INCA-M."**

Lines 354 – 366 – This section jumps around a lot across the different panels of Figure 6 and is difficult to follow. For example,

First, the comparison between the different runs shows a better correlation in the reference simulation in the UT, possibly suggesting that the effects from biomass burning and lightning emissions on ozone production are realistically distributed in space. As expected, no change is observed in the LS for this metric, since the higher amounts of ozone in the LS increase the NOx threshold necessary to trigger a net ozone production (e.g. Hegglin et al., 2006).'

From the second sentence it is not clear what metric is being discussed when it is stated 'no change is observed in the LS for this metric'.
It has been replaced by a more explicit formulation:
*"As expected, no change* **in the ozone correlation** *is observed in the LS,* **[...]"**

Lines 367 – 368 – 'the shorter and lesser NOy sampling does not lead to strong differences.' might be clearer as 'the shorter period of time and sparser measurements of NOy does not lead to strong differences.'
The formulation suggested by R1 has been substituted to the previous one.

Lines 373 – 375 – I would agree with the statement 'It is likely that the influence of biomass burning on the LS is overestimated because of an excessive exchange between the troposphere and the stratosphere.' The change in bias in CO in the LS is quite surprising.

Line 384 – I can deduce what is meant by 'barycentre' but it is not a correct word. 'mean pressure of the measurements' maybe?
The phrase has been shortened, but made clearer:
*"The mean pressures on the right axis have been added in order to identify changes in mean altitude measurements."*

---

## Author Comment (AC2)

**Authors' response to Reviewer 2**

We are thankful to both reviewers for their positive and accurate feedback on our study, and for the improvements they helped us make to this paper.

The responses are organized as follows: the reviewer's comment is in blue, our answers are in black, and the changes proposed for the revised manuscript are in italic (black for modified sentences, grey for unchanged sentences that have been pasted here in order to remind the context).

This manuscript provides climatological chemical composition information regarding the upper troposphere (UT) and lower stratosphere (LS), based on a fairly long-term compilation of in situ aircraft data (from IAGOS, as described in some detail), focusing on a gridded/averaged version of these data sets for O3, CO, NOy, and H2O, and a comparison versus a similarly gridded chemical model (LMDZ-OR-INCA). The comparisons entail annual mean mapped comparisons and seasonal comparisons as well, scatter plots, along with Taylor diagrams to highlight certain aspects of the model runs, which include simulations with no biomass burning or with no lightning NOx impacts; this helps to evaluate certain contribution factors regarding these climatologies in the UTLS. Although Taylor diagrams are not always liked by everyone, and there are a fair number of such plots, I found that for this purpose, they do add some useful information regarding the overall model characteristics and top-level fits to the data. The separation between the UT and LS regions, and the "all data regions together (UTLS)" was found to be useful as well. This manuscript is appropriate for publication in ACP, after some minor adjustments, minor, overall.

My main request (or query, at least) has to do with whether slightly more information could be mentioned regarding the water vapour lower stratospheric data and related comparisons, for both the data and model implications. The authors point out that the drier conditions in this region mean that the IAGOS H2O sensor is likely to have a wet bias, as mentioned on page 16 before section 3.2.1, and the filter they apply to try to ameliorate this issue leads to a low bias, still, for the model versus the IAGOS H2O data in this region (midlatitude lower stratosphere). Is it clear (at all) whether the model or the data might be more "at fault" for the LS discrepancies? If the data set is really expected to come with a high bias, can the morphology of the differences still be of some use or are we just stuck not being able to deduce much, except in the UT region, where the comparisons are viewed as among the best (between O3, CO, NOy, and H2O). Maybe there is a way to make some comparisons with other data sets (e.g., ballonborne data sets from Boulder, or elsewhere, and/or with satellite data sets, see for example the Read et al., 2022 article in AMT, on UT H2O intercomparisons, if this might be useful). I will accept that this probably goes beyond the current goals/expectations for this manuscript, if no

further comments can be currently made about the disagreements between the model and the data for H2O. However, since there might be indications of an overestimated tropopause/stratosphere exchange, with high model CO biases, for example, in the lower stratosphere, would this not also lead to larger model values of H2O in the LS? Yet, the model clearly underestimates the IAGOS H2O values there. It is unfortunate if nothing more robust can even be mentioned (or possibly even speculated about), but if this is the case, this will need to await better measurements and/or validation exercises for the IAGOS LS H2O data. The authors could still (try to) mention something slightly more informative, even if they have been leary not to do so, so far.
Ignoring the above, especially if it seems too challenging of a topic, given the current state of the LS H2O data, I do have a number of quite minor wording comments, with a few suggestions for clarifications, as listed below.
After these matters are considered/answered, I see no reason not to recommend this manuscript for publication, as the comparisons offer some nice insights to the community regarding these species in the UTLS, and they highlight mostly good comparisons with the model that is presented. As the authors point out, it would indeed be useful for future studies to pursue multi-model comparisons with IAGOS data, in order to enable some better discrimination among the various models regarding their representation of various processes (e.g., transport, convection, biomass burning, lightning NOx sources, and ultimately longer-term variations in the UTLS).

We thank Reviewer 2 (R2) for his/her suggestions and for the careful review, which is of great help for the quality of this manuscript.

Concerning the water vapour climatologies in the LS, we agree that we would need to clarify whether the model underestimates or not the mixing ratios. Other in-situ measurements can be used, but it is unfortunately beyond the scope of this paper. However, there is an ongoing work (Konjari et al., in preparation) adjusting the IAGOS-Core measurements of H2O, using the comparison with the measurements from IAGOS-CARIBIC, which product could be processed for the next models assessments against IAGOS. The bias linked with the sensor is generally less than 20 %, and is thus not sufficient to explain the differences between IAGOS-DM and INCA-M. Instead, it is likely that the filtering method not only lowers the mean altitude of the measurements, but also selects particularly moist air masses, too transient to be captured by the model.

The following figure shows the annual climatologies of water vapour in the LS, from both the IAGOS-DM and the INCA-M products, and both with (first two panels) and without (last two panels) the 20 % filtering regarding the occurrence of very dry air masses. It is clear that this filtering process increases the difference between the model and the observations, leading to a greater negative "bias". However, as it can

be seen in the two panels on the right, it does not explain why the INCA-M product shows drier climatologies than IAGOS-DM.

[Figure]

It does not exclude that the model effectively underestimates LS water vapour despite an excess of cross-tropopause mixing, since the ice-supersaturation is not implemented in the current version, which can induce an overestimated sedimentation of ice particles.

In the following, all R2's minor comments have been addressed. For a given comment, we use the answer "done" to notify that we agree with R2 and that we apply the suggestions/corrections exactly as it is proposed.

Mostly technical and a few minor comments/questions:
- Line 1, change "the global" to "global"
Done.

- L2, I suggest: toward an improved understanding of the chemical composition in this region. This composition is regularly sampled...
Done.

- L7, I would suggest adding "(sub-sampled)" after "masked", for clarity.
Done.

- L12, I suggest: There are opposite model CO biases in the UT and the LS, which suggests that...
Done.

- L14,15, need to reword the 2nd half of this sentence so it can be properly understood. Please clarify the meaning ("and the ozone latitudinal variations" is not

tied to the rest of the sentence, and the CO part is also not tied that well, and does not directly follow from a good simulation for water vapour). Maybe use a semi-colon after "water vapour" and then write what you want to convey for the 2nd part of the sentence.

Following the suggestions from R2, the phrase has been reformulated into:

*"In the tropics, the upper-tropospheric climatologies are remarkably well simulated for water vapour*. They also show realistic CO peaks due to biomass burning in the most convective systems, and the ozone latitudinal variations are well correlated between the observations and the model."

- L17, depends on location and season. The present study demonstrates that...
Done.

- L18, assessment of global model simulations...
Done.

- L19, "chemistry climate or chemistry transport models."
Done.

- L23, with varying strength
Done.

- L26, ozone absorbs most of the energetic ultraviolet...
Done.

- L27, the air's oxidizing capacity
Done.

- L29, regarding the formation and life cycle of cirrus clouds, whose large radiative forcing still carry a large uncertainty...
Done.

- L31, thus increasing the CH4 lifetime.
Done.

- L33, classified as essential climate variables
Done.

- L34, NOx gets converted back and forth into its reservoir species ...
Done.

- L37, chemical species and their radiative forcings
Done.

- L40, uncertainties in dynamical processes.
Done.

- L42, NOy also provides information
Done.

- L44, chemical destruction, NOy can also...
Done.

- L45, mass origins... CO, on one hand, and O3 and NOy on the other hand, are more
...
Done.

- L48, The assessment of CCM or CTM simulations relies on comparisons with
Done.

- L49, few observations are suited for diagnosing... and few can account for UTLS
vertical
Done.

- L55, ", but they are too sparse..."
Done.

- L57, provide frequent and large-scale
Done.

- L58, programs have highlighted large-scale features since the 1970s; these
programs include TROZ..., GASP..., and more recently, NOXAR..., [all?] with an
observation period of four years or less.
Done.

- L62, Since more than two decades ago, the In-service... has provided...
Done.

- L68, took advantage of the whole IAGOS database.
Done.

- L80, of the impact of lightning and...
Done.

- L81, the present study goes further into the development and application ...
Done.

- L84, O3, CO, but also H2O...
Done.

- L89, biomass burning to the modelled...
Done.

- L90, account for differences in the definitions of seasons and in the mean tropopause altitude.
Done.

- L95, Its predecessors, MOZAIC... and CARIBIC..., relied on the same principle.
Done.

- L101, Since the merging...respective databases are referred to as ...
Done.

- L102, hereafter, with an approach validated by...
Done.

- L104, measured with an ultraviolet ...
Done.

- L112, sensor ranges from 5 to 70% RHL...
Done.

- L118, the case of the frost-point...
Done.

- L123, you need to define the acronym "ORCHIDEE" here.
Done.

- L130, aerosols scheme includes a total of 123 tracers and 22 aerosol tracers.
Done.

- L133, please provide a reference (or more) to clarify this statement regarding past comparisons.
Two recent studies are given in example now, in order to refer to comparisons in the UTLS region of LMDZ-INCA with ozonesondes (Terrenoire et al., 2022, ACP), with the IASI satellite-borne sensor (Dufour et al., 2021, ACP) and with airborne MOZAIC measurements (Brunner et al., 2003 and 2005, ACP).

- L137, mostly dealing with chlorine and bromine chemistry, along with 66 gas-phase

...
Done.

Done.

Done.

According to R1 and R2 comments, some clarifications have been brought to this point:
*"a sampled grid point is considered as upper-tropospheric if its PV is lower than 2 PVU, if it is not the first gridpoint below the 2 PVU isosurface, and [...]"*

Done.

Done.

Done.

Done.

Done.

Done.

Done.

We agree that the formulation can lead to confusion, notably because "partial column" is not an appropriate term for a simple vertical average. The first phrases of this paragraph have been changed, in order to clarify the explanation:

*"A time series of seasonal means is calculated for each grid point, and then averaged throughout the years. The mean yearly climatologies are then defined as the average between the four seasonal climatologies. In the end, the 3-dimensional climatologies are averaged vertically throughout the cruise altitude levels."*

Please clarify how broad the "typical" regions are in the UT and in the LS (separately), regarding the IAGOS measurements used in this work; maybe pointing to Figure 7 is sufficient, but please help the reader make this connection, if possible.

According to R2, we calculated a geographical distribution of the vertical range represented by each vertical average (based on the annual climatologies, in the mid-latitudes, and for ozone measurements). The results are summarized in the table below, in terms of relative frequencies (in %):

|  | < 1 km | 1 – 2 km | 2 – 3 km | 3 – 4 km |
|---|---|---|---|---|
| UTLS | 21 | 28 | 40 | 11 |
| LS | 30 | 41 | 28 | 1 |
| UT | 32 | 39 | 22 | 8 |

In the separated UT and LS, the observations spread more frequently (~40 %) on a vertical range between 1 and 2 km, but they also spread frequently on a vertical range less than 1 km (~30 %) and between 2 and 3 km (22 and 28 %, respectively). As expected in the non-separated UTLS, the observations spread on wider vertical ranges, more frequently between 2 and 3 km.

The manuscript has been modified as follows (L253 – L257):

*" **Also**, it must be kept in mind that the UTLS layer is not solely the merging of the UT and the LS, since it also comprises the vertical range between 2 and 3 PVU that separates the two layers. **Last, the altitude range of cruise measurements varies geographically as well. In the northern extra-tropics, the vertical range of the ozone measurements varies mostly between less than 1 km up to 3 km, with a maximum frequency (~ 40 %) between 1 and 2 km for the separated UT and LS, and between 2 and 3 km for the non-separated UTLS."***

- L206, change "meridian" to "meridional" [and at several other places in the manuscript].
Done.

- L218, a factor of 4 for the...
Done.

- L227, change "this couple of metrics is defined as" to "these two metrics are

defined as:"
Done.

- L236, We use these metrics to evaluate the reference simulation.
Done.

- L244, these Figures 1-4 should have been better explained in terms of how you obtain average VMR values based on "column measurements" mentioned earlier. Even if a past reference explains this better, providing a brief summary here would be useful.
One brief reminder has been added in this paragraph:
*"Ozone, CO, NOy and water vapour yearly distributions in the UTLS, UT and LS are shown in Figs. 1 – 4 respectively, and their corresponding seasonal averages are available in Supplementary Material. **They represent vertical averages through the cruise altitudes.** Showing the results both [...]"*

- L249, sampling relative to the tropopause...geographically, as a result of tropopause and cruise altitude variations.
Done.

- L255, the minimum in the western equatorial ...
Done.

- L258, like the good model...
Done.

- L260, characterized by a smaller geographical variability...
Done.

- L262, moderate positive CO bias...
Done.

- L263, NOy is characterized by discrepancies...
Done.

- L266, a noticeable minimum East of Central America.
Done.

- L267, NOy tends to be overestimated
Done.

- L268, NOy is underestimated.  meridian --> meridional
Done.

- L270, maximum in H2O [or CO or?] above the most convective regions...
Done.

- L271, Atlantic Ocean, as well as the collocated ozone minimum. This [H2O or O3?] feature is due to...
Done.

- L290, visible changes in the MNMB or in the correlation.
Done.

- L291, but at most, it can be interpreted as an upper limit.
Done.

- L299, For a given species, we note that there are high correlations [between what and what? model and data?] in the layer where the mixing ratios are at a maximum (LS for ozone,...).
Done.

- Table 2 is good, although you probably do not need that many digits for MBNB and FGE (-0.086 could be written as -0.09 for example) [it is not critical to change this however, but it is not needed or significant enough in my view; same for Table A1].
Agreeing with R2's comment, we reduced the amount of significant numbers in all the MNMB and FGE values.

- L335, water vapour mixing ratios at low latitudes...
Done.

- L340, nitrogen have poorer scores, with lower correlation coefficients and an underestimated geographical variability.
In this quote, we kept the "more underestimated" formulation, in order to keep comparing the CO/NOy variability with that of the other species.

- L345, The Taylor diagrams in Fig. 6 present a synthesis of ...
Done.

- L355, is it not also true that the longer chemical lifetime for ozone in the LS implies more of a dynamical control than a chemical one [this is not exactly the same as just stating that O3 abundances are larger in this region]?
It is true that a longer ozone lifetime will tend to homogenize the ozone mixing ratio inside a given layer. The impact of the LNOx injected from the troposphere into the lower stratosphere on ozone may thus be less local, which can also lead to an unchanged correlation between model and observations in ozone in the LS.

The paragraph has been modified as follows:

*"As expected, no change in the ozone correlation is observed in the LS. * **One possible reason is that** *the higher amounts of ozone in the LS increase the NOx threshold necessary to trigger a net ozone production (e.g. Hegglin et al., 2006).* **Another possible explanation is that ozone has a longer lifetime in the LS than in the troposphere: the impact of LNOx injections into the LS might thus be more homogeneous than in the UT, which is consistent with a less sensitive ozone geographical variability to lightning in the LS."**

- L356, This is consistent with the fact...

Done.

- L359, please clarify which r value corresponds to which run.

It has been clarified:

*"(r=0.67* **for the reference run,** *compared to r=0.53* **for the run without lightning***)"*

- L361, change "lightnings" to "lightning"

Done.

- L363, 'layer, such as too much convection.' This overestimated tropospheric influence could be consistent (it sems) with the overestimated H2O amounts, it seems to me; you may want to comment about this.

We added a last phrase to this paragraph:

*"[...] such as too much convection*, **which could also explain the water vapour positive bias in the UT."**

- L364, more important in the run without LNOx.... In the "no LNOx" run, model ozone has a significant negative bias (from -15 to -20% ... in annual means), and model NOy has a small bias (...), while model CO is increased and shows a 10-50% positive bias. [is this how one should write/read this sentence?]

The sentence has been reformulated as follows, in order to clarify while keeping the information on the variations between the reference run and the sensitivity runs:

*"[...] in the run without LNO$_x$ than without biomass burning.* **In the former run, ozone is decreased and shows an important negative bias (from -15 to -20 % throughout the layers, in annual means), NO$_y$ is decreased and shows a small bias (between -10 and 0 %), while CO is increased up to a 10 – 50 % positive bias due to decreased OH concentrations."**

It also seems that an issue with model transport (or mixing) could affect the model NOy values [but not clear if this would be consistent with too much LS model H2O].

According to this point, we added the following phrase at the end of the paragraph:

*"There are several possible explanations, including a lack of nitric acid (HNO$_3$) loss by scavenging in the troposphere and/or heterogeneous reactions.* **The lack of scavenging**

*combined with the overestimation of the cross-tropopause exchanges would be consistent with the non-lightning NO$_y$ overestimation in all the layers."*

[but not clear if this would be consistent with too much LS model H2O].
It is true, but in our case, we can hardly interpret the LS H$_2$O in IAGOS-DM. As explained previously in this document, the filtering applied to the H2O observations might select only anomalous humid conditions in the LS that are too transient to be reproduced by the model. It is probably the main source of difference between IAGOS-DM and INCA-M, more than the measurement biases themselves.

- L372, I suggest rewording as follows [or clarifying]: "In the run with no biomass burning, we observe decreases in CO, and the annual model CO bias changes from -5% to -15% in the UT, from 30 to 15% in the LS, and from 15 to 0% in the UTLS. Surprisingly, the impact of biomass burning is not negligible in the LS, especially in the summer.
We thank R2 for this suggestion of rewording, that we adopted.

- L374, again, an excessive net model transport between troposphere and stratosphere goes along with high values of H2O...
We agree, but as in a previous response, the H$_2$O climatologies shown in this paper cannot be interpreted physically.

- L376, "This suggests that this season maximizes..."
Done.

- L384, measurements' barycentre . They can be associated with significant...
According to R1, we reworded this sentence:
*"The mean pressures on the right axis have been added in order to identify changes in mean altitude measurements."*

- L387, are thus difficult to interpret [if this is what you mean]
This was indeed the right meaning. It has been clarified in the manuscript.

- L392, Their small number of occurrences indicates that seasonal mean...
Done.

- L393, which provides more confidence regarding the representativeness of the NOy measurements in the context of the whole ozone measurement period.

- L401, This is consistent with the peak... [and can probably delete "calculated"]
Done.

- L407, above the South American tropics...
Done.

- L408, association with the start...
The word "association" has finally been removed from this phrase:
*"The southern CO maximum that we observe here is thus due to the start of an enhanced convective activity while biomass burning emissions are still intense."*

- L423, This is consistent with enhanced ozone...
Done.

- L424, as shown by Barret et al. (2016), and as suggested by the important seasonal...
Done.

 [but what does "important seasonal O3/CO ratio mean? please clarify, does this mean "large seasonal O3/CO ratio variations highlighted by Cohen et al. (2018)?]. I would also break up this long sentence: "; this was also confirmed by measurements from the ...HALO during the HALO-ESMVal campaign...
The point on the $O_3$/CO ratio has been reworded as follows:
"large seasonal $O_3$/CO ratio highlighted in this region by Cohen et al. (2018)"

- L428, stratospheric intrusions to the northern side
Done.

- L430, They linked such important photochemical activity with a combination of uplifted precursors...
Done.

- L434, Good consistency between... obsrvations is visible for ozone...
Done.

- L436, well the H2O maximum at 5-10...
Done.

- L437, along the northern tropics...
Done.

- L440, has difficulties in reproducing the extremely...
Done.

- L453, above Africa from December to March and from June to October...
Done.

- L457, The simulated NOy profiles underestimate the observed meridional

variability.
Done.

- L458, Above Africa, NOy is almost systematically underestimated by the model...hemisphere, but the NOy comparisons show a general consistency in the northern hemisphere.
Done.

- L459, an important positive NOy model bias during the Asian summer...
Done.

- L468, simulations show some strong variability.
Done.

- L469, lightning NOx in CO destruction...
Done.

- L473, notably between the opposite subtropics, probably reflecting an underestimated modeled [?] interhemispheric transport.
The point was not to assess the model abilities in reproducing the interhemispheric transport, because this comparison does not provide any clue about the rightness of this model transport pathway.

- L476, CO local maxima simulated...May are not visible
Done.

- L492, consistent with the literature...
Done.

- L496, is associated with local biomass..., , as [is?] a significant part of the peak
Done.

- L497, In contrast, the observed CO maximum during April-May...is rather associated with other sources.
Done.

- L500, two regions is mainly caused by biomass burning.
Done.

- L501, consistent with the literature.
Done.

- L503, reduces wildfires via enhanced precipitation.
Done.

- L505, This study presents an assessment of...
Done.

- L521, underestimated by the model in the UT, and model CO shows a positive bias in...
Done.

- L522, overestimation in the model's extra-tropical cross-tropopause mixing [or net transport?]. Again, such a model feature might affect the large values of H2O (?).
We agree that water vapour should also be impacted by this transport feature, but it is hard for us to interpret the H2O averages in the LS.

- L523, This is likely linked...
Done.

- L525, tracks; aircraft can indeed play a significant role in the NOy levels.
Done.

- L531, I would also strongly recommend that you add a sentence (or two) here regarding the large values of modeled H2O in the LS.
We fully agree with this Reviewer 2's recommendation. The text has been modified as follows:
*"Last, concerning water vapour in the LS, the IAGOS-Core humidity sensor was initially designed for tropospheric air masses. Though a filter has been applied in an attempt to exclude most of the measurements likely to overestimate the humidity, the corresponding climatologies in the LS shown in this study **still** cannot be used to assess the model simulation. One possible explanation is that the filtering method makes the IAGOS $H_2O$ mean values only representative of particularly moist conditions (on a sub-daily scale), thus increasing substantially the difference with the model output."*

- L534, able to accurately represent the mean...
Done.

- L538, this sentence is not clear enough; you quote 3 numbers (25, 30, 45 ppb CO) but the previous sentence seems to mention only two seasons (D-M and J-O) unless you need to add the convective season above South America (please specify which months), or clarify the 3 numbers and their relationship to the previous sentence more carefully.
The months have been added for the convective season above South America (September – November).

- L539, you should probably (briefly) explain in this section the CO sink from lightning

emissions, for readers that might not read the whole paper...

A brief explanation has been added to this sentence, as shown below:

*"In these cases, the model attributes respectively 25 ppb, 30 ppb and 45 ppb of the CO peaks to biomass burning, and attributes between 10 and 20 ppb of the CO sink to lightning emissions.* **The latter enhances the CO destruction by increasing the ozone production, which in turn increases the OH production.***"*

- L542, where enhanced CO is attributed to biomass burning peaks.

Done.

- L544, inconsistencies in model ozone and CO with respect to...

Done.

- L547, is likely to enhance the model skills for NOy and ozone.

Done.

- L549, daily output or the model monthly output

Done.

- L551, in the framework of the second phase

Done.

- L553, Other potential applications include ... scales and for seasonal scales, as well as interannual..., possibly also allowing for source apportionment regarding the observed features.

 Done.

- Figure 5 (3rd line in caption), fit described in the top-left corner... [rather than on the top-left corner]. Change "the amount of grid points" to "the number of grid points".

Done.

- Figure 6, the assessment of the yearly climatologies...

Done.